# Coding of chromatic spatial contrast by macaque V1 neurons

**Abhishek De[1,2], Gregory D Horwitz[2]***

[1]Graduate Program in Neuroscience, University of Washington, Seattle, United States; [2]Department of Physiology and Biophysics, Washington National Primate Research Center, University of Washington, Seattle, United States

**Abstract** Color perception relies on comparisons between adjacent lights, but how the brain performs these comparisons is poorly understood. To elucidate the underlying mechanisms, we recorded spiking responses of individual V1 neurons in macaque monkeys to pairs of stimuli within the classical receptive field (RF). We estimated the spatial-chromatic RF of each neuron and then presented customized colored edges using a closed-loop technique. We found that many double-opponent (DO) cells, which have spatially and chromatically opponent RFs, responded to chromatic contrast as a weighted sum, akin to how other V1 neurons responded to luminance contrast. Yet other neurons integrated chromatic signals nonlinearly, confirming that linear signal integration is not an obligate property of V1 neurons. The functional similarity of cone-opponent DO cells and cone non-opponent simple cells suggests that these two groups may share a common underlying circuitry, promotes the construction of image-computable models for full-color image representation, and sheds new light on V1 complex cells.

## Editor's evaluation

There was a strong agreement that the work fills an important gap in the understanding of spatial integration of color signals in primate visual cortex and that the use of novel stimulus paradigms to study color responses in alert macaque provided important new information. There was also strong support for the analytical approaches used to categorize simple and double opponent neurons as well as appreciation for the quality of the discussion of the results.

*For correspondence:
ghorwitz@uw.edu

**Competing interest:** The authors declare that no competing interests exist.

## Introduction

Color depends on the spectral composition and spatial organization of lights (*Brown and MacLeod, 1997*; *Kraft and Brainard, 1999*; *Monnier and Shevell, 2003*; *Shevell and Monnier, 2005*; *Xian and Shevell, 2004*). The joint spectral and spatial statistics of lights in natural scenes are critical for object recognition, visual memory, and scene segmentation (*Fine et al., 2003*; *Spence et al., 2006*; *Wurm et al., 1993*). Despite its importance for these critical functions, however, the spatial integration of spectral signals by neurons in the visual system is poorly understood.

Some V1 neurons combine signals from all cone photoreceptor types (from which they receive any input at all) with the same sign. Many of these cone non-opponent neurons have receptive fields (RFs) consisting of side-by-side ON and OFF subfields, rendering them sensitive to luminance contrast of a particular orientation at a particular location in their RFs. Other V1 neurons combine signals from multiple cone types with opposite sign and are therefore cone-opponent. A subset of V1 cone-opponent neurons, called double-opponent (DO) cells, are both cone-opponent and spatially opponent, rendering them sensitive to chromatic edges of particular orientations at particular locations in their RFs (*Conway, 2001*; *De and Horwitz, 2021a*; *Johnson et al., 2008*). V1 is the first stage of

the primate visual system at which DO cells exist (*Hubel and Wiesel, 1968*; *Livingstone and Hubel, 1984*).

V1 is also the first stage of the primate visual system that contains simple cells (*Hubel and Wiesel, 1968*). These cells, by definition, respond to a drifting achromatic sinusoidal grating of appropriate spatial- and temporal-frequency with spike rate modulation that exceeds the unmodulated response (*Skottun et al., 1991*). Applied to macaque V1 neurons, this definition likely includes many DO cells (simple cells were originally characterized in cat, an animal with few if any cone-opponent V1 neurons). We use the term 'simple cell' in this report to refer to phase-sensitive, orientation-tuned macaque V1 neurons that are *not cone-opponent*, so as to distinguish them from DO cells. Cone non-opponent simple cells have much in common with DO cells (*De and Horwitz, 2021a*). A major goal of the current study was to determine whether these two cell types integrate visual information across their RFs similarly.

Classically defined simple cells combine light information across their RFs approximately as a weighted sum (*Carandini et al., 1997*; *DeAngelis et al., 1993*; *Hubel and Wiesel, 1959*; *Movshon et al., 1978b*). These neurons can therefore be thought of as linear filters that operate on the retinal image. This insight propelled scientific progress in at least three ways. First, it facilitated the construction of image-computable models of achromatic image representation (*Adelson and Bergen, 1991*; *Marr and Hildreth, 1980*; *Mehrotra et al., 1992*). Second, it shed light on cortical circuitry: simple cells receive excitation (push) and inhibition (pull) with opposite spatial tuning, and this appears to be a critical step for establishing linearity in the face of nonlinear input from the lateral geniculate nucleus (LGN) (*Ferster, 1988*; *Ferster and Miller, 2000*; *Hirsch et al., 1998*; *McLaughlin, 2000*). Third, it provided an essential building block for quantitative models of V1 complex cells (*Adelson and Bergen, 1985*; *Movshon et al., 1978a*), neurons in higher-order cortical areas (*Cadieu et al., 2007*; *Freeman et al., 2013*; *Okazawa et al., 2015*; *Rust et al., 2006*; *Simoncelli and Heeger, 1998*; *Willmore et al., 2010*), and a variety of psychophysical phenomena (*Adelson and Bergen, 1985*; *Beaudot and Mullen, 2006*; *Graham, 2011*; *Malik and Perona, 1990*; *Moreland and Boynton, 2017*; *Wilson et al., 1997*). Currently, all of these advances have been restricted to the achromatic domain. A similar quantitative understanding of image representation in the chromatic domain is lacking and is necessary to extend these advances to natural, full-color images.

The nature of the spatial antagonism implemented by DO cells has important implications for their contributions to vision. For example, consider a hypothetical DO cell that compares cone-opponent signals between the left and right halves of its RF (*Figure 1A*). If the cell integrates signals linearly and with equal weight, then it is well suited for extracting vertical chromatic edges (*Figure 1B–D*). In this case, the excitation produced by a preferred light in one half of the RF can be canceled by the same light in the neighboring half. On the other hand, if the cell integrates signals nonlinearly, for example, by weighting contrast increments more heavily than contrast decrements, it would encode both edges and surfaces (*Figure 1E and F*). In this case, a light in one half of the RF would fail to cancel the same light appearing in the neighboring half.

In this study, we measured the spatial integration of visual signals by individual V1 neurons in awake, fixating monkeys. Neurons can combine signals in many ways, but we focus on linear combinations because of their theoretical significance. Linearity is a mathematical ideal that is never fully realized by neurons, so we do not ask whether neurons integrate visual information across their RFs linearly but instead quantify how well a linear model describes spatial integration relative to a more flexible model. Models were fit to data collected using a closed-loop stimulus generation technique (*Bölinger and Gollisch, 2012*; *Horwitz and Hass, 2012*). The closed-loop technique involved the real-time construction of optimally oriented and positioned edges that varied in color but drove identical spike count responses. An advantage of this approach is that it is robust to static output nonlinearities.

Using this technique, we found that many DO cells responded linearly to differences in cone-opponent signals across their RFs, in quantitive similarity to how other V1 neurons responded to spatial differences in luminance. Nevertheless, a significant proportion of spatially opponent neurons combined signals across their RFs nonlinearly and in ways that could neither be attributed to a static output nonlinearity nor to a nonlinear combination of signals from linear subfields. These results contribute to our understanding of cone signal combination in V1 and reveal a population of DO cells that combine signals linearly across their RFs. We speculate that the linear DO cells provide input

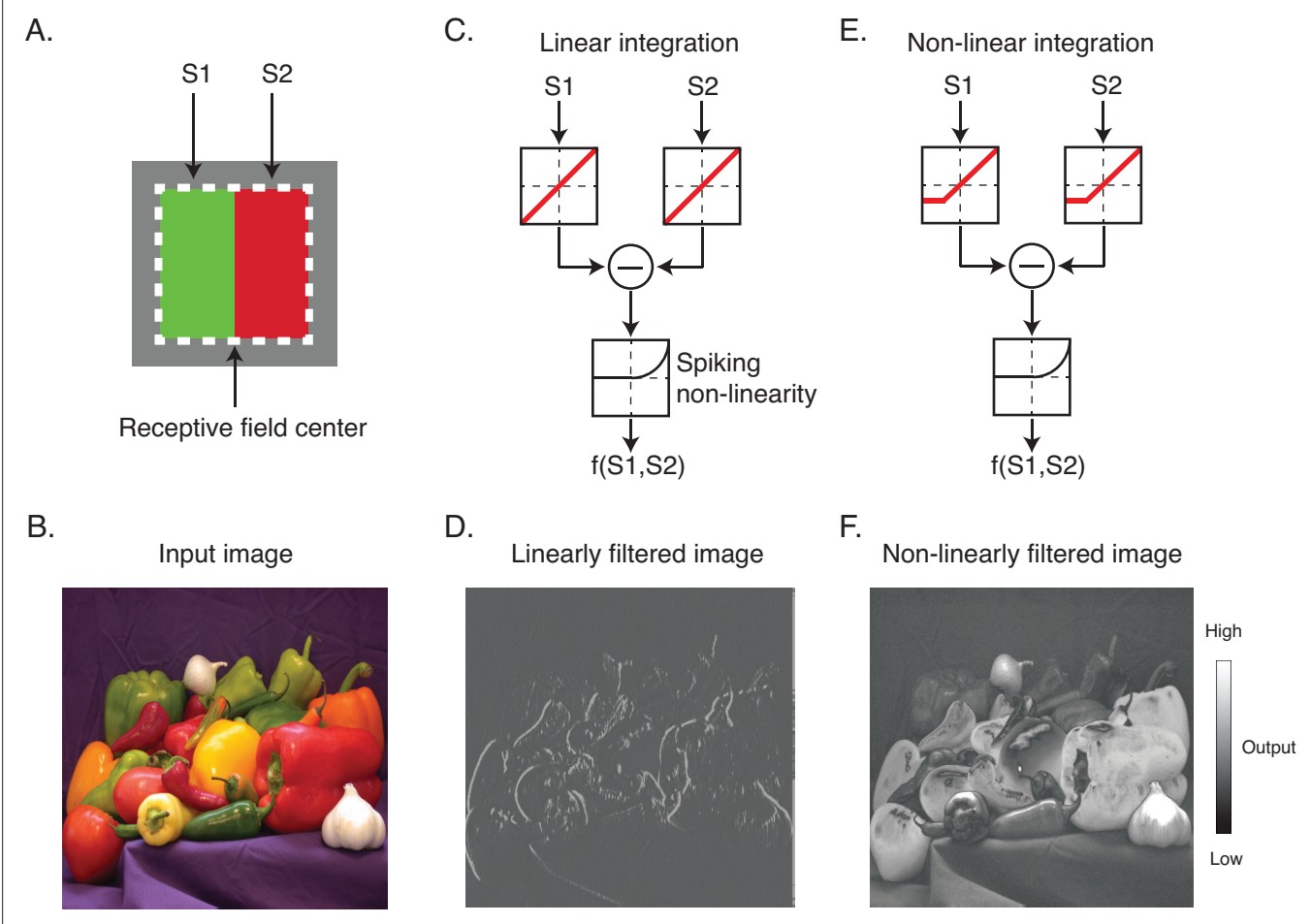

**Figure 1.** Linear and nonlinear image filtering. (**A**) A hypothetical V1 DO cell that is excited by a red light on one side of its receptive field (RF) and a green light on the other. S1 and S2 represent stimulation of the two subfields of the RF. (**B**) An example natural image. (**C**) A linear spatial filter that sums the drive from each subfield and generates a response via a spiking nonlinearity. Note that the drive from each subfield is combined linearly before being transformed by the spiking nonlinearity. (**D**) Linearly filtered image using filter in (**C**). (**E**) A nonlinear spatial filter that partially rectifies S1 and S2 prior to summation. (**F**) Nonlinearly filtered image using filter in (**E**). DO, double-opponent.

to complex cells, a proposal that explains several previous results and provides new insight into the functional role of color-sensitive V1 complex cells.

## Results

We analyzed the spiking responses of 98 V1 neurons from two awake, fixating male macaque monkeys (69 from Monkey 1, 29 from Monkey 2). Each neuron was visually stimulated with pixel white noise: a 10×10 grid of square, 0.2° pixels, each of which changed randomly and independently on every screen refresh. Data were analyzed by spike-triggered averaging to identify a pair of functionally distinct subfields within the classical RF. Visual stimulation was then targeted to these subfields to characterize their individual and joint contributions to the neuron's firing rate.

### RF characterization

Spike-triggered averages (STAs) of some neurons resembled uniform blobs, indicating invariant spectral sensitivity across the RF. Other STAs were spatially structured, for example, consisting of a set of yellow pixels adjacent to a set of blue pixels (*Figure 2A*). These structured STAs are a signature of neurons that compare spectral information across space. If an STA did not reveal at least two distinct subfields, the neuron was passed over for data collection.

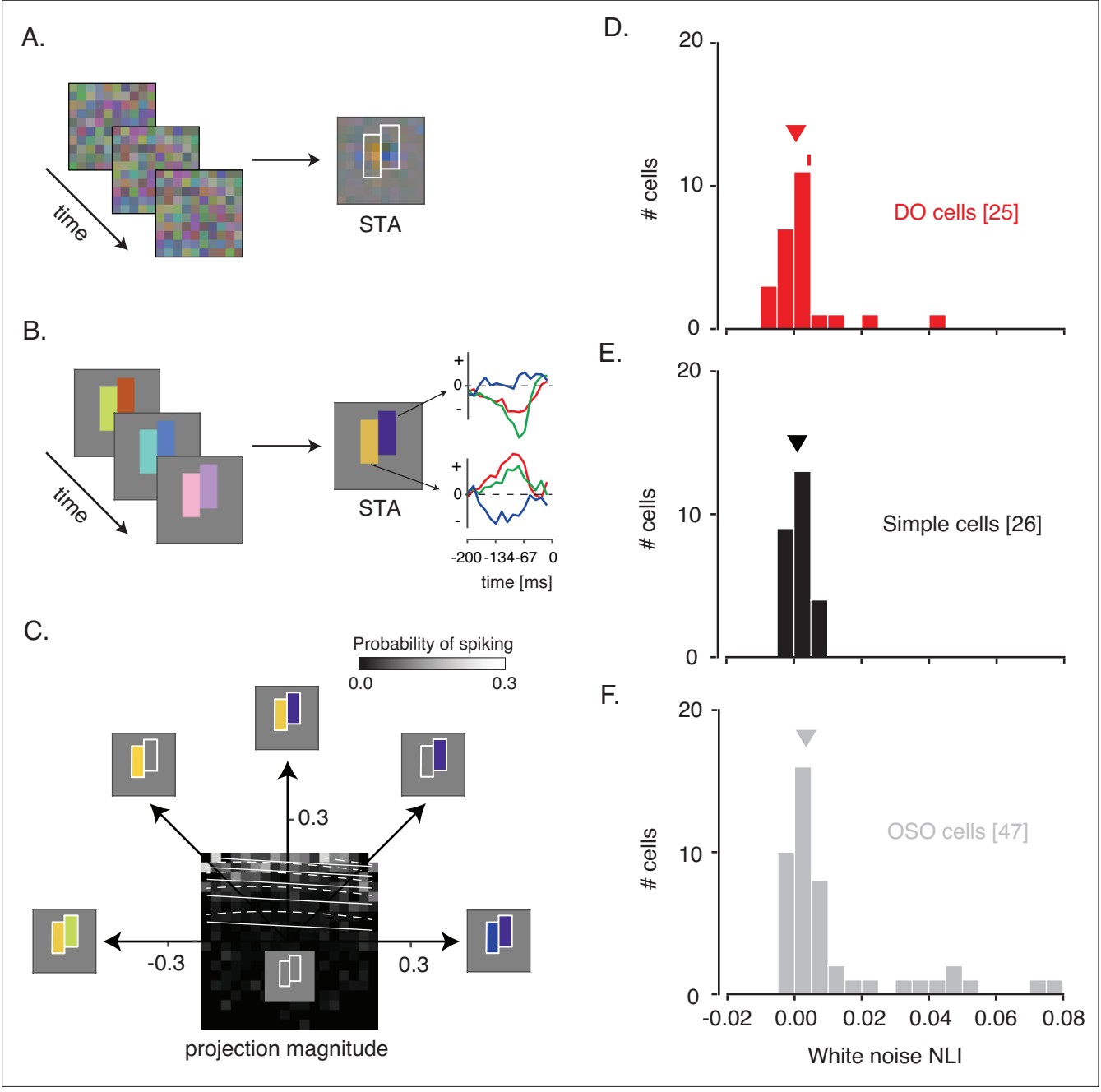

**Figure 2.** White noise analysis of RF structure and spatial integration. (**A**) Pixel white noise stimulus (left), spike-triggered average (STA; right). Two sets of contiguous pixels were yoked to create two hyperpixels, each of which stimulated one RF subfield (white outlines). (**B**) Hyperpixel white noise stimulus (left) and STA at the peak frame (middle). Red, green, and blue curves (right) represent the average red, green, and blue phosphor intensities, relative to background, as a function of time before a spike. (**C**) A firing rate map for the example DO cell. The probability of spiking (grayscale) is plotted as a function of projection magnitudes of the hyperpixel stimulus onto the right and left halves of the STA (the 45° and 135° directions, respectively). Solid and dashed white lines are contours of constant spiking probability from GLM and GQM fits to the data, respectively. The firing rate map is binned to facilitate visualization, but models were fit to unbinned data. (**D**) Histogram of white noise NLIs for DO cells. The NLI of the example neuron is marked with a tick, and the median is marked with a triangle. (**E**) Same as (**D**) but for simple cells. (**F**) Same as (**D**) but for the other spatially opponent cells. DO, double-opponent; GLM, generalized linear model; GQM, generalized quadratic model; NLI, nonlinearity index; RF, receptive field.

The online version of this article includes the following figure supplement(s) for figure 2:

**Figure supplement 1.** Comparison of the effective stimulus contrast across the three phases of the experiment.

**Figure supplement 2.** Comparison of firing rates across the three phases of the experiment.

*Figure 2 continued on next page*

*Figure 2 continued*

**Figure supplement 3.** Cone weights for DO cells (**A**) and simple cells (**B**) derived from each hyperpixel STA.

**Figure supplement 4.** Overview of hyperpixel white noise nonlinearity index calculation.

**Figure supplement 5.** Relationship between white noise NLI and the eigenvalue corresponding to the largest principal component (PC1) for two simulated neurons.

A major goal of these experiments was to characterize spatial integration within the RFs of DO cells. The pixel white noise stimulus is ill-suited for achieving this goal because it modulates V1 neurons weakly, and nonlinear spatial integration might appear linear in response to small perturbations. To drive the neurons more effectively, we customized the stimulus for each neuron studied. From the pixel white noise STA, we identified two contiguous groups of pixels, each covering an RF subfield, and yoked each group into a 'hyperpixel' (*Figure 2B*). White noise modulation of the two hyperpixels stimulated the two subfields strongly and independently (*Figure 2—figure supplement 1*), driving a wide range of firing rates (*Figure 2—figure supplement 2*).

On the basis of responses to the hyperpixel white noise stimulus, neurons with spatially opponent RFs were classified into three categories. The largest category consisted of neurons with a significant excitatory stimulus feature that was distinct from the STA (n=35, see Material and methods). Some of these neurons may have been complex cells, a classification that is inconsistent with the classical definitions of simple cells and DO cells. These neurons were therefore given the designation 'other spatially opponent' (OSO). For the 63 cells not classified as OSO by this criterion, cone weights were computed and analyzed. An additional 12 neurons with cone weights insufficient for a decisive cone-opponent or non-opponent classification were also classified as OSO. The remaining 51 neurons were divided into DO cells and simple cells on their basis of cone-opponency. Altogether, 26 neurons were classified as simple cells, 25 were classified as DO cells, and 47 were classified as OSO cells.

The two sides of the hyperpixel STAs from simple cells and DO cells were complementary, or nearly so. This complementarity was evident via a comparison between the background-subtracted red, green, and blue phosphor intensities at the two hyperpixels (Pearson's r=–0.94±0.11 (mean ± SD) for simple cells and –0.76±0.23 for DO cells), and between cone weights derived from these RGB values (*Figure 2—figure supplement 3*). The results of these analyses are color space-dependent, but the fact that similar results were obtained in the physiologically relevant space of cone contrast and the device-specific space in which sampling errors are independent across color channels, attests to the spatial opponency of simple cells and DO cells in our data set.

## Measuring spatial integration using hyperpixel white noise

We quantified interactions between RF subfields using an approach similar to one used previously to study interactions between stimulus features that trigger spikes in complex cells (*Rust et al., 2005*; *Touryan et al., 2002*). In these previous studies, white noise stimuli were projected onto the plane spanned by the first and second principal components of the spike-triggering stimuli. Similarly, we projected the hyperpixel white noise stimuli onto the two halves of the STA (see Materials and methods: Hyperpixel white noise analysis of signal combination across subfields). These two projection values reveal how similar the stimulus was to the two halves of the hyperpixel STA: the larger the projection, the more of the STA is present in the stimulus. We visualized a firing rate map by binning stimulus projections and calculating the proportion within each bin that drove a spike (*Figure 2C*). The probability of spiking increased with the stimulus projection onto individual RF subfields, and it rose more steeply when both projections increased together.

To analyze spatial integration between RF subfields, we fit the (unbinned) data with a generalized linear model (GLM) and a generalized quadratic model (GQM) (see Materials and methods: Hyperpixel white noise analysis of signal combination across subfields; *Figure 2—figure supplement 4A*). We quantified the ability of the two models to classify stimuli as spike-triggering or not using receiver operating characteristic (ROC) analysis (*Figure 2—figure supplement 4B*; *Green and Swets, 1966*). Spike-triggering stimuli were defined as 15-frame-long segments of the white noise movie for which a spike occurred on the final frame.

Classification error rates of the GLM and GQM were summarized with a white noise nonlinearity index (NLI) (see Materials and methods: White noise nonlinearity index). A white noise NLI<0 indicates

that the GLM provides more accurate predictions than the GQM. An NLI>0 indicates that the GQM provides more accurate predictions than the GLM. An NLI of 0 occurs if the GLM and GQM make identical predictions, which can occur because the GLM is a special case of the GQM with three parameters set to zero. Because of these extra parameters, the GQM always fits the training data as well or better than the GLM. To compare the two models fairly, we tested the model on data that had been held out from the fitting using tenfold cross-validation (*Browne, 2000*).

NLIs differed across the three cell types (median white noise NLI for DO cells=0.0005, simple cells=0.0009, OSO cells=0.0034; p=0.02, Kruskal-Wallis test; *Figure 2D–F*). Comparison between simple and DO cells revealed no significant difference between them (p=0.99, Mann-Whitney U-test). To the contrary, NLIs of simple and DO cells were both lower than those of the OSO cells (p<0.05, Mann-Whitney U-test). This latter result is not entirely surprising given that any neuron that had a significant excitatory stimulus feature distinct from the STA was classified as OSO (see Materials and methods: Spike-triggered covariance). However, this criterion does not force the result (*Figure 2— figure supplement 5*). In short, neurons that we classified as simple or DO combined signals across space with a similar degree of linearity and more linearly than other V1 neurons that also had spatially structured STAs.

In interpreting these data, it is important to note that a lack of evidence for a difference between simple and DO cells is not evidence that a difference does not exist. This experiment probed neurons with low-contrast, rapidly modulated stimuli (*Figure 2—figure supplement 1*). The possibility remains that differences between simple and DO cells become evident when they are tested with stimuli of higher contrast or longer duration. We tested this possibility in *Phase 3* of our experimental protocol, as described below.

## Measuring spatial integration using the isoresponse method

For each neuron, we found a collection of stimuli that evoked the same response. Each stimulus was spatially identical to the hyperpixel STA, but the contrast of the two hyperpixels was adjusted according to the algorithm described in Materials and methods: Contrast staircase procedure. Negative contrasts were allowed.

To appreciate the necessity of this technique, consider a classical alternative. A classic test of linearity is to present one stimulus at the RF of a recorded neuron, then another, and then both together. If the response to the combined stimulus does not equal the sum of responses to the two components, the neuron is not linear. However, this test is sensitive to nonlinearities that are logically distinct from the linearity of spatial summation and are present in otherwise linear neurons (e.g., spike firing thresholds and saturating contrast-response functions). An alternative approach is to find a collection of stimuli that evoke the same response from an isolated neuron and analyze these stimuli to identify the features they share. This approach has been used previously to study signal integration in the salamander retina and locust auditory receptor cells (*Bölinger and Gollisch, 2012*; *Gollisch et al., 2002*). It has also been used previously to analyze the linearity of signal integration across cone types by individual macaque V1 neurons (*Horwitz and Hass, 2012*), but it has not been used previously to analyze the linearity of signal integration across a V1 RF.

If a neuron combines cone-contrast signals linearly across its RF, then stimuli that drive the same response will lie on lines when represented in the stimulus space shown in *Figure 2C* or any other stimulus space that is a linear transformation away from it (e.g., cone excitation difference or opponent modulation spaces [*Brainard, 1996*]). If the stimuli lie on a curve instead of a line, the hypothesis of linear spatial summation can be rejected. This approach makes no assumptions about static nonlinearities downstream of spatial integration whereas the GLM and GQM assumed a logistic function. It also does not assume linearity of cone signal integration within individual RF subfields. This assumption was used in the GLM and GQM analyses to reduce RGB values from 15-frame-long stimulus movie fragments to two stimulus projections.

In *Figure 3A*, each point represents a stimulus, distance from the origin represents contrast relative to the background, and angle represents contrast between the two sides of the stimulus. Within this plane, a search was performed to find physically distinct stimuli that evoked the same neuronal response. Angles were selected pseudo-randomly, and distances were titrated by a staircase procedure until a target firing rate was achieved (*Figure 3—figure supplement 1*). To mitigate the impact of spontaneous activity on the staircase procedure, the target firing rate for each neuron exceeded

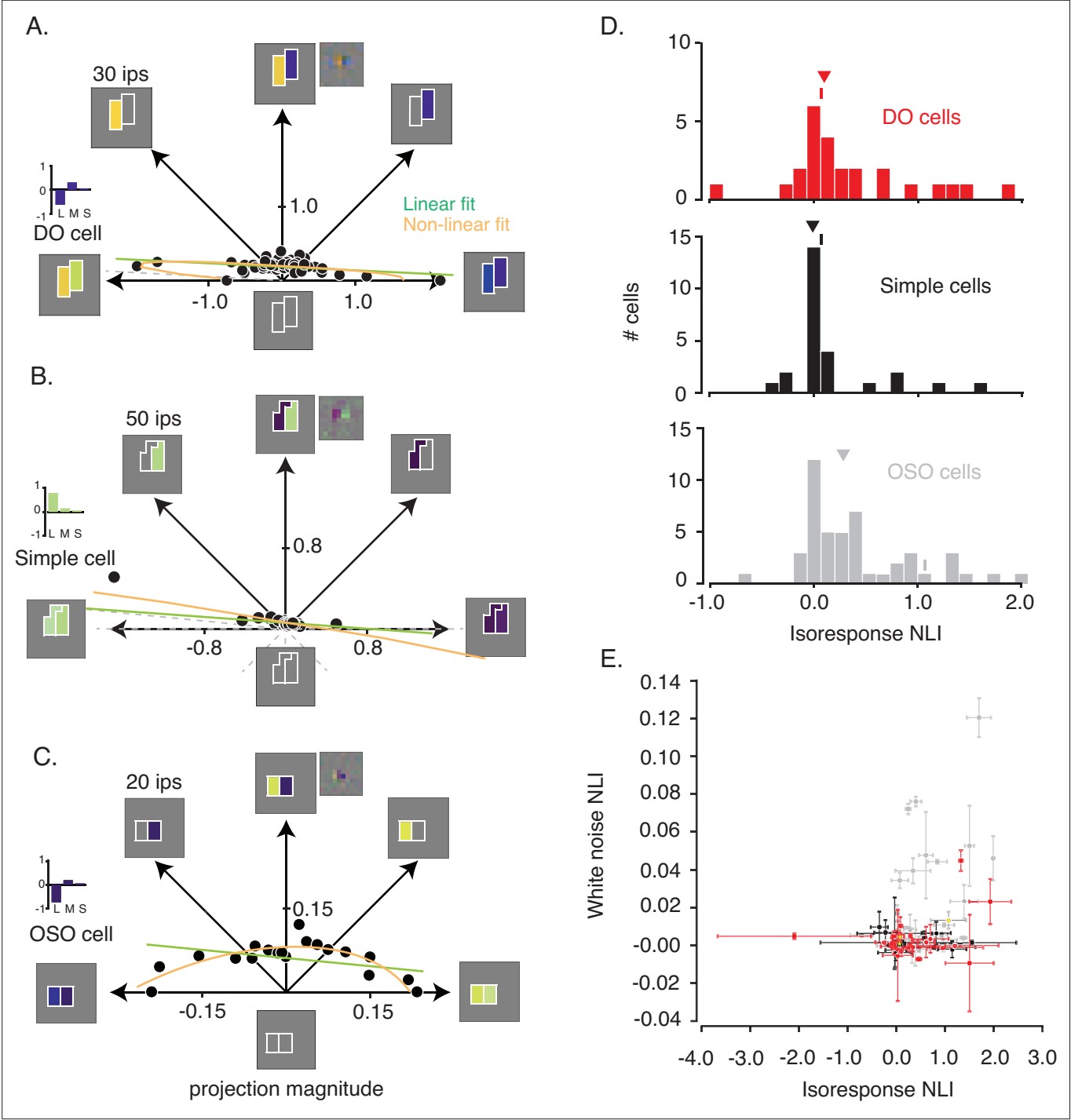

**Figure 3.** Analysis of isoresponse contours. (**A**) Data from the example DO cell shown in *Figure 2A–C*. Dots indicate staircase terminations (target firing rate=30 ips) and gray dashed lines indicate staircases that exceeded the monitor gamut. Linear (green) and nonlinear (orange) fits to the data are similar. Pixel STA (to the right of the hyperpixel STA) and cone weights derived from the color weighting functions (bar plot) are also shown. (**B**) Same as (**A**) but for a simple cell (target firing rate=50 ips). L+M spectral sensitivity manifests as bright green (ON) or dark purple (OFF) when probed with RGB white noise (*Chichilnisky and Kalmar, 2002*). (**C**) Same as (**A**) for a cell that was spatially opponent but neither simple nor DO (target firing rate=20 ips). (**D**) Histogram of isoresponse NLIs. NLIs of example neurons are marked with ticks, and medians are marked with triangles. (**E**) Scatter plot of isoresponse NLIs and white noise NLIs. Example neurons are marked with yellow asterisks. Error bars were obtained via a jackknife procedure (see Materials and methods). DO, double-opponent; NLI, non-linearity index; STA, spike-triggered average.

The online version of this article includes the following figure supplement(s) for figure 3:

*Figure 3 continued on next page*

*Figure 3 continued*

**Figure supplement 1.** An example staircase from the closed-loop procedure used to study the DO cell in *Figure 2A*.

**Figure supplement 2.** Distribution of baseline and target firing rates during *Phase 3* for DO (red), simple (black), and OSO (gray) cells.

**Figure supplement 3.** Zoomed-in isoresponse contours and staircase terminations from *Figure 3A–B*.

**Figure supplement 4.** Cross-validated errors from linear (absicca) and nonlinear (ordinate) fits to data from *Phase 3* for DO cells (red), simple cells (black), and OSO cells (gray).

**Figure supplement 5.** White noise and isoresponse NLIs from neurons reclassified using alternative cone weight criteria.

the 90th percentile of the baseline firing rate distribution, and for 95/98 neurons, the target firing rate exceeded the 95th percentile (*Figure 3—figure supplement 2*). Target firing rates did not differ across cell types (p=0.63, Kruskal-Wallis test).

For some neurons, staircase termination points lay close to a line when plotted in the stimulus space (*Figure 3A–B* & *Figure 3—figure supplement 3*). This result shows that the excitation produced by a preferred light in one part of the RF can be canceled by an anti-preferred light in a neighboring part with a fixed constant of proportionality over the entire gamut of our video display. This cancelation is consistent with linearity of spatial integration (*Figure 1A*) and not with differential sensitivity to contrast increments and decrements (*Figure 1B*). However, not all neurons behaved this way. For some neurons, staircase termination points lay on a curve (*Figure 3C*), consistent with nonlinear spatial integration.

To determine quantitatively whether a line or a curve provided the better description of the staircase termination points, we compared linear and quadratic models fits for each neuron (see Materials and methods: Evaluating model fits to staircase termination points; *Figure 3—figure supplement 4*). We defined an isoresponse nonlinearity index (isoresponse NLI) similarly to the white noise NLI defined previously (see Materials and methods: Evaluating model fits to staircase termination points). An isoresponse NLI of 0 indicates that the linear and quadratic models made equally accurate response predictions, NLI<0 indicates that the linear model predicted responses more accurately than the quadratic model, and NLI>0 indicates that the quadratic model predicted responses more accurately than the linear model. Cross-validation ensured that the quadratic model did not achieve greater prediction accuracy simply by virtue of having more parameters.

Isoresponse NLIs of DO cells and simple cells were close to zero and did not differ significantly (median isoresponse NLI for DO cells=0.1007, median isoresponse NLI for simple cells=–0.0097; p=0.14, Mann-Whitney U-test; *Figure 3D*). In contrast, NLIs were greater for OSO neurons (median isoresponse NLI=0.2822, p=0.02, Kruskal-Wallis test). We conclude that neurons that we classified as simple cells and DO cells are similarly linear over the range that we were able to test given the limits of our display and that they are more linear than other neurons in V1.

Isoresponse NLIs (from *Phase 3* of the experiment) were positively correlated with white noise NLIs (from *Phase 2* of the experiment) (r=0.30, p=0.001, Spearman's correlation; *Figure 3E*). This correlation was driven primarily by OSO cells (r=0.41, p=0.004,) and not by DO (r=0.01, p=0.95,) or simple cells (r=–0.19, p=0.34). Even for the OSO cells, however, this correlation was far from perfect. Many neurons had white noise NLIs near 0 but isoresponse NLIs>>0. The reason for this discrepancy is unclear but is consistent with the higher contrasts used in *Phase 3* engaging nonlinear mechanisms that were not engaged in *Phase 2*. However, we cannot rule out the possibility that isoresponse NLIs were inflated by nonstationarity in firing rate combined with the sequential measurement procedure used in *Phase 3*, or by other differences in the data collection or analysis procedures.

## Measuring signal integration within individual subfields

An appreciable fraction of neurons in our data set, principally those in the OSO category, were poorly described by the linear model. We asked whether the nonlinear model shown in *Figure 1E* described these neurons more accurately. Under this model, signals from the three types of cone photoreceptors combine linearly within each RF subfield and are then transformed nonlinearly prior to spatial integration.

We tested the assumption of linear combination within a subfield by regressing spikes recorded during *Phase 2* onto linear and nonlinear combinations of the three display primaries within each subfield. If signals from multiple cone types combine linearly within each RF subfield, the influence of

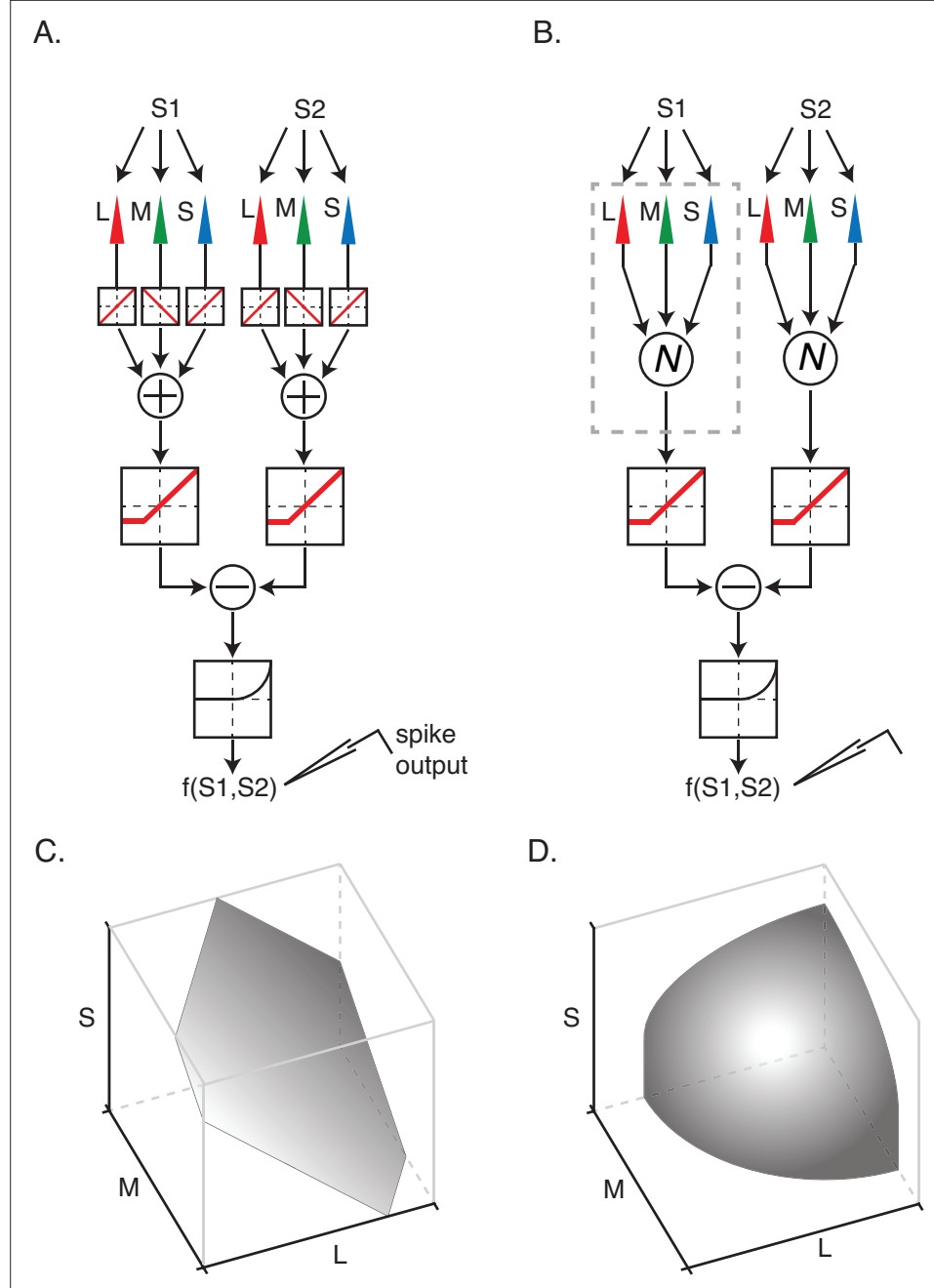

**Figure 4.** Two models of nonlinear spatial integration. (**A**) A model in which signals from the three types of cone photoreceptor are integrated linearly within a subfield. The combined cone signals within each of the subunits are transformed nonlinearly and then summed to generate a response, f(S1,S2). (**B**) A model in which cone signals are nonlinearly transformed within a subfield prior to linear combination. (**C**) An isoresponse surface of firing probability is plotted as a function of cone contrasts for stimulation of a single subfield. (**D**) Same as (**C**) but for nonlinear integration model in (**B**). The isoresponse surface is curved in (**D**) and planar in (**C**) because signal combination within subfields is nonlinear in (**B**) and linear in (**A**).

that subfield on the spiking response can be summarized by a weighted sum of red, green, and blue primary intensities.

In this analysis, we treat the influence of each subfield as additive noise on the signal produced by the other. This approach is justified by the facts that the two hyperpixels in *Phase 2* modulated

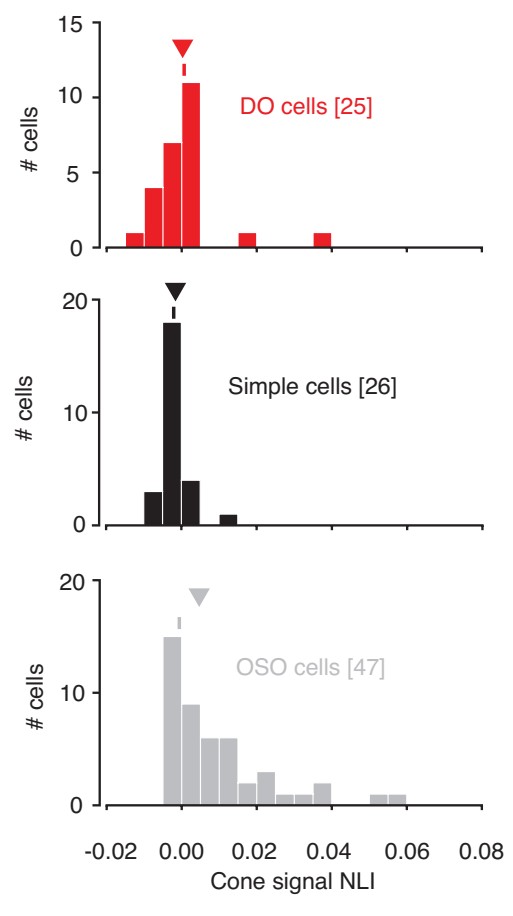

**Figure 5.** Within-subunit nonlinearity indices (NLIs) were calculated from GLMs and GQMs that predicted spikes on the basis of the RGB components of individual hyperpixels. NLIs of example neurons in *Figure 3* are marked with ticks, and medians are marked with triangles. GLM, generalized linear model; GQM, generalized quadratic model.

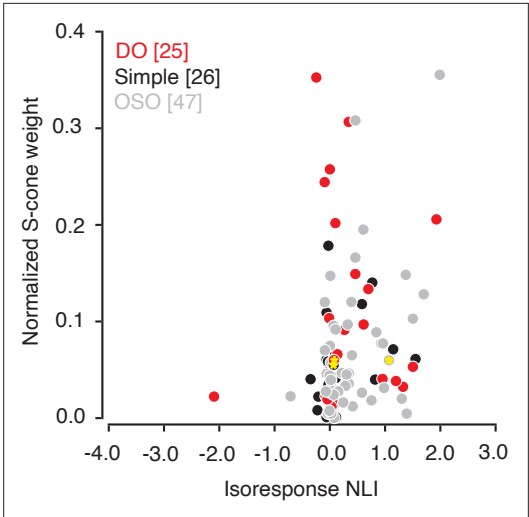

**Figure 6.** Relationship between S-cone input and signal integration across subfields. Scatterplot of normalized S-cone weight magnitude and isoresponse NLI. Data from example neurons are marked with yellow asterisks Note that the cone weights for OSO neurons are added for the sake of completeness but must be interpreted with caution as many of the these neurons combine cone signals nonlinearly. Cone-weights are meaningful only for linear neurons. NLI, nonlinearity index; OSO, other spatially opponent.

independently, and under the model, each subfield makes an additive contribution to the firing rate (*Figure 4A*).

For each neuron, we fit four models (two models × two hyperpixels) to the hyperpixel white noise data: a GLM and a GQM that predicted spiking responses as a function of the background-subtracted RGBs at each hyperpixel (*Figure 4*). We computed model performance using ROC analysis and quantified the difference in performance of the GLM and GQM with an NLI (see Materials and methods: Hyperpixel white noise analysis of signal combination within subfields). Model performance was averaged across hyperpixels to provide a single 'within-subfield' NLI per neuron.

Within-subfield NLIs differed across the three cell types (p<0.0001, Kruskal-Wallis test; *Figure 5*). On average, within-subfield NLIs were higher for OSO cells than for simple or DO cells (median within-subfield NLI for simple cells=–0.0016, for DO cells=–0.0003, for OSO cells=0.0047, p<0.0001, Kruskal-Wallis test). This result extends to individual RF subfields the observation that simple cells and DO cells combine signals with a similar degree of linearity and shows that the nonlinear signal integration exhibited by many OSO cells is poorly described as resulting from a nonlinear combination of two linear RF subfields.

## Relationship between S-cone input and signal integration across and within subfields

Many V1 neurons that are driven strongly by modulations of the S-cones are poorly described by a linear model (*Horwitz et al., 2005*). Consistent with this observation, we found that the absolute value of the normalized S-cone weight derived from the hyperpixel STA was correlated with isoresponse NLI

(r=0.23, p=0.02, Spearman's correlation; *Figure 6*). However, S-cone weight did not correlate significantly with the other two NLIs (white noise and within-subfield, both r<0.2, p>0.05).

## Discussion

A fundamental goal of visual neuroscience is to characterize the transformation from lights to the spiking responses of individual neurons. In this study, we characterized the spatial integration of light by neurons in area V1 of macaques with two types of visual white noise stimulation, closed-loop isoresponse measurements, and statistical model comparisons. We found that some DO cells integrate signals across their RFs linearly, like cone non-opponent simple cells do and in contrast to other V1 neurons. We did not find strong evidence for V1 neurons that pool information linearly across cone photoreceptor types within a subfield and then nonlinearly across subfields.

Below, we compare our results to those from previous studies and discuss how our results are affected by cell classification criteria and the monkeys' eye movements. We then discuss the implications of our results for the circuitry that underlies DO and simple cells and how these two cell types may contribute to downstream image processing. We conclude with speculations on parallels between the processing of color and other stimulus features in V1 by complex cells.

### Relationship to previous work

Linearity in the visual system is 'a rare and (apparently) prized commodity in neural signal processing' (*Shapley, 2009*). The linearity of V1 simple cells is not an accident of random convergence of LGN afferents but rather the product of specialized excitatory and inhibitory circuitry (*Ferster, 1988*; *Hirsch et al., 1998*; *Tolhurst and Dean, 1990*). The discovery that V1 simple cells combine signals linearly across their RFs contributed to scientific progress in many ways. It provided a bridge between neurophysiology and the fields of psychophysics and computer vision. It provided guidance for how to characterize neuronal stimulus tuning efficiently. It served as a basis for more elaborate models; all V1 neurons exhibit some degree of nonlinearity, but the linear model remains a cornerstone of even nonlinear V1 models (*Carandini, 2006*; *Carandini et al., 2005*).

Nearly all quantitative studies of V1 spatial RF properties have used achromatic stimuli (but see *Conway, 2001*; *Conway et al., 2002*; *Conway and Livingstone, 2006*; *Johnson et al., 2001*; *Johnson et al., 2008*). Extending quantitative RF mapping to color is complicated by the curse of dimensionality. As the dimensionality of the color space increases from 1-D (achromatic) to 3-D (full color), the number of possible spatial combinations grows exponentially. Classic workarounds include the use of gratings, which have a highly constrained spatial structure and/or cone isolating stimuli, which are most useful for analyzing neurons that combine signals linearly across cone type. Our solution was to map the RF of each neuron with 3-D (full color) white noise and then customize spatial patterns on the basis of these maps.

Three previous studies investigated spatial integration by DO cells. Using the 2-bar interaction technique, *Conway, 2001* found that most color-sensitive V1 neurons responded maximally when a pair of bars that modulated different cone types appeared side-by-side within the RF. This maximal response exceeded the response to either bar in isolation, consistent with linearity of spatial summation as well other models (e.g., those that include an expansive nonlinearity after linear spatial summation). At least one cell was found to integrate signals nonlinearly (*Conway, 2001*). *Conway and Livingstone, 2006* measured the responses of DO cells to cone-isolating stimuli at individual RF locations. Most of the DO cells they studied showed clear signs of push-pull inhibition, which is correlated with linearity in V1 (*Ferster, 1994*; *Ferster and Miller, 2000*; *Hirsch et al., 1998*; *Tolhurst and Dean, 1990*). Our findings extend these results by demonstrating the linearity of spatial integration directly through simultaneous stimulation of functionally distinct RF subfields.

Most other recent studies of DO cells used cone-isolating stimuli and limited mixtures of those stimuli, which cannot completely reveal interactions among cone types (*Conway, 2001*; *Johnson et al., 2001*, *Johnson et al., 2004*; *Conway and Livingstone, 2006*; *Johnson et al., 2008*). In contrast, we used a stimulus set that modulated all three cone types together in a variety of proportions. In further distinction from other studies, we stimulated DO cells with colored edges to confirm the spatial and spectral sensitivity inferred from the STAs.

## Cell categorization criteria

We classified neurons as OSO (other spatially opponent) if their hyperpixel STAs indicated spatial opponency and the first principal component (PC1) of hyperpixel white noise stimuli preceding spikes (PC1) was larger than expected by chance. This latter criterion was necessary to satisfy the assumptions underlying the conversion of the STA to cone weights, and it ensured that the neurons surviving this filter did not exhibit full-wave rectified responses, which would be inconsistent with the classical definitions of simple cells and DO cells.

The PC1 criterion was based on the outcome of a statistical hypothesis test and not the bimodality of a distribution (no evidence of bimodality was observed). Indeed, we suspect that DO cells and at least some OSO cells lie on a continuum. Neurons in both categories compare cone-opponent (and possibly non-opponent) signals across space and may comprise multiple distinct subtypes. We further note that some OSO cells had small NLIs, which is consistent with linear spatial integration, and some DO cells had large NLIs, consistent with nonlinear integration. This lack of a clear distinction between these cell types could be related to the fact that the PC1 and NLI are sensitive to overlapping but distinct types of nonlinearity, these measures depend on the number of spikes in different ways (the PC1 criterion was based on a statistical test, and the NLI was not), and nonlinearities that are clear with high-contrast, long-duration stimuli are not always detectable with white noise (*Tanabe and Cumming, 2008*).

Some cells that we classified as OSO would likely have been classified as DO in other studies (e.g., *Conway and Livingstone, 2006*; *Johnson et al., 2008*). In our study, as in a previous one, neurons were found with STAs consistent with a DO classification but PC1s consistent with a complex cell classification (*Horwitz et al., 2007*). Whether these neurons are more usefully classified as nonlinear DO cells, partially rectified complex cells, or something else entirely is an important question that is partly physiological and partly semantic. In any case, a major finding of this study is that a population of DO cells combines cone-opponent signals across their RFs approximately as linearly as simple cells combine non-opponent signals, a result that stands despite the existence of other V1 cells that integrate signals nonlinearly.

The cone weight criteria for inclusion into the simple cell and DO cell categories were asymmetric because the variability in estimated L- and M-cone weights is greater for non-opponent cells than for opponent cells (*Horwitz et al., 2007*). Reclassifying DO, simple, and OSO cells with different cone weight criteria did not change the main results of this study (*Figure 3—figure supplement 5*).

## Eye movements

The monkeys, although rewarded for fixating, made small eye movements that moved the stimuli relative to the RF. These eye movements blurred STAs and inflated PC1s (for cells with spatially opponent RFs), but these effects were modest because the eye movements were small relative to the stimulus elements. Across experiments, the average standard deviation of horizontal and vertical eye positions (including measurement error) was 0.13°, and the largest standard deviation across all experiments was 0.18° (for reference, the size of each pixel in *Phase 1* was 0.2°). Neurons with RF subfields too small to produce a spatially opponent STA were passed over for data collection. Importantly for the conclusions of this study, neither the horizontal (H) nor vertical (V) standard deviation varied significantly across cell category (Kruskal-Wallis tests, p<0.1) or correlated with isoresponse NLI (Spearman rank correlation coefficients, r=–0.03 (H), –0.18 (V), p>0.05). The fact that we were able, for many neurons, to cancel the effect of an increase in preferred color on one side of the RF with a decrease in preferred color on the other side during *Phase* 3 of the experimental protocol attests to the stability of the image on the retina relative to the sizes of the RFs we studied. We conclude that eye movements affected the measurements but are unlikely to have contributed substantively to the main conclusions of this study.

## Neural circuitry underlying DO and simple cells

The spectral sensitivity of a V1 neuron is determined by the spectral sensitivity of its excitatory and inhibitory afferents. A cell that is excited by L-ON and M-ON afferents in one part of its RF, and by L-OFF and M-OFF afferents in another, is spatially opponent but not cone-opponent. Spatial linearity in such a cell could be implemented via spectrally matched push-pull inhibition: L-OFF and M-OFF inhibition from the ON subfield and L-ON and M-ON inhibition from the OFF subfield. Pooling the

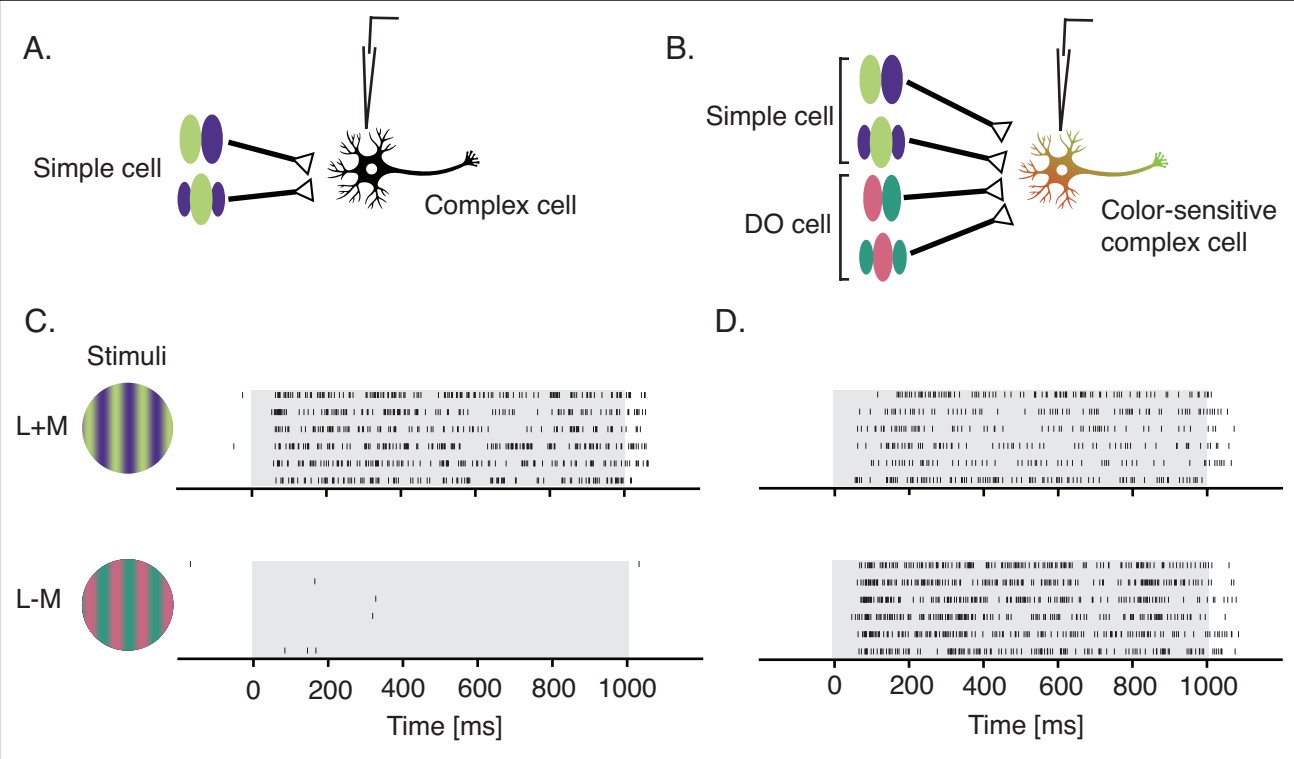

**Figure 7.** Proposed signal convergence of simple cells and DO cells onto complex cells. (**A**) A hypothetical complex cell receiving input from simple cells with overlapping odd- and even-symmetric receptive fields. (**B**) A hypothetical color-sensitive complex cell receiving input from simple and DO cells. (**C**) Response of a complex cell to a drifting sinusoidal grating that modulated L- and M-cones at 3 Hz with identical contrast in phase (top) and in anti-phase (bottom). (**D**) Same as (**C**) but for a color-sensitive complex cell. Gray overlays indicate stimulus duration. Both of these neurons were recorded before the hyperpixel white noise was developed and probably would have been passed over for data collection in this study because of their phase insensitivity. DO, double-opponent.

same afferent signals to produce cone-opponency (e.g., pairing L-OFF with M-ON and L-ON with M-OFF) with otherwise identical circuitry would produce a linear L-M DO cell.

S-cone signals, which are prevalent in DO cells and, to a lesser degree in simple cells, defy this simple explanation. S-cone ON and OFF pathways in the retina and LGN are highly asymmetric with regard to the L- and M-cone signals they carry (*Tailby et al., 2008*; *Wool et al., 2019*). One possibility is that these asymmetries contribute to the nonlinearity of S-cone-dominated V1 neurons.

## Contributions to downstream processing

Simple cells are thought to provide the dominant input to complex cells (*Alonso and Martinez, 1998*; *Hubel and Wiesel, 1962*). Under a standard model, complex cells pool signals from simple cells with overlapping RFs and shared preferred orientation (*Figure 7A*). One possibility is that some complex cells also receive input from DO cells (*Figure 7B*). We speculate that complex cells receiving (cone non-opponent) simple cell input only are luminance-sensitive (*Figure 7C*) whereas those that receive input from simple and DO cells are both color- and luminance-sensitive (*Figure 7D*). This conjecture is consistent with the observation that the preferred orientation of color-sensitive complex cells is maintained across color directions (*Johnson et al., 2001*). It is also consistent with the observation that color-sensitive complex cells have multiple preferred color directions by STC analysis (*Horwitz et al., 2007*) and no null directions in cone-contrast space (*Horwitz and Hass, 2012*). Complex cells that are insensitive to chromatic modulation abound (e.g., *Figure 7C*), whereas those that are insensitive to luminance modulation are exceedingly rare, if present (*Horwitz et al., 2007*).

Alternatively, the chromatic sensitivity of color-sensitive complex cells could arise entirely from DO cell inputs (*Michael, 1978*), which have been reported to be similarly responsive to chromatic and luminance contrast (*Johnson et al., 2001*; *Johnson et al., 2004*; *Johnson et al., 2008*). Yet another possibility is that it arises directly from LGN inputs. This explanation is consistent with the direct

koniocellular projections to the cytochrome oxidase blobs in area V1 (*Xiao, 2014*), but is difficult to reconcile with the sharp orientation tuning and spatial phase-invariance of some color-sensitive complex cells across color directions. These visual tuning properties would seem to require a phase- and orientation-selective upstream intermediate or else elaborate dendritic computations.

### Analogous neural coding of color and stereopsis

The stereotyped microcircuitry of area V1 contributes to vision for form, color, depth, and motion. These distinct visual modalities have distinct computational demands, but V1 circuits may contribute to each via a small set of operations that process different signals in similar ways. Parallels in the processing of binocular disparity and motion direction in V1 are well established (*Adelson and Bergen, 1991*). We speculate that color and stereopsis have heretofore unappreciated parallels and that models of binocular disparity tuning may provide a useful guide for the study of cone-opponent and non-opponent signal combinations in V1.

A fundamental component of binocular V1 models are simple cells that linearly sum signals from the two eyes (*Anzai et al., 1999*; *Ohzawa and Freeman, 1986*; *Read and Cumming, 2003*). Analogous building blocks in the color domain are simple cells and DO cells that sum cone non-opponent and cone-opponent signals with a similar degree of linearity. Binocular simple cells are thought to provide input to binocular complex cells that implement a binocular energy calculation (*Read and Cumming, 2003*). An analogous convergence of simple cell and DO cell outputs would implement a color energy calculation (*Barnett et al., 2021*; *Horwitz and Hass, 2012*).

The binocular energy model, while extremely successful in describing complex cell responses, fails to account for the attenuation of responses to anti-correlated signals between the two eyes (*Cumming and Parker, 1997*). This specialization of real V1 cells is thought to reflect the statistics of natural inputs to the visual system (*Haefner and Cumming, 2008*). Under natural viewing conditions, binocularly correlated patterns are more common than anti-correlated patterns, and V1 neurons are specialized to encode them.

A parallel phenomenon may exist in the domain of color. Under natural viewing conditions, luminance and chromatic spatial gradients tend to be aligned, and their alignment (or misalignment) carries important information regarding the physical sources of the gradients. Edges between different materials under fixed illumination produce in-phase luminance and chromatic modulations, whereas uncorrelated variations in illumination and pigmentation, such as produced by curved three-dimensional objects of non-uniform reflectance, produce out-of-phase modulations (*Cavanagh, 1991*; *Kingdom, 2003*; *Kunsberg et al., 2018*). The energy model produces phase-invariant responses and so does not account for these specializations. Whether real V1 neurons respond in accordance with the energy model or show enhanced responses to natural alignments between chromatic and luminance modulations is an important and unanswered question.

## Materials and methods

**Key resources table**

| Reagent type (species) or resource | Designation | Source or reference | Identifiers | Additional information |
|---|---|---|---|---|
| Strain, strain background (*Macaca mulatta*, male) | Monkey | Washington National Primate Research Center | | Rhesus monkey |
| Software, algorithm | Matlab | Mathworks | https://www.mathworks.com/products/matlab.html RRID:SCR_001622 | |
| Software, algorithm | Sort Client | Plexon | http://www.plexon.com RRID:SCR_003170 | |
| Software, algorithm | Offline Sorter | Plexon | http://www.plexon.com RRID:SCR_000012 | |

## Contact for resource sharing

Further information and requests for resources should be directed to and will be fulfilled by the Lead Contact, Gregory D. Horwitz (ghorwitz@u.washington.edu). Data and analysis code are available at https://github.com/horwitzlab/Chromatic_spatial_contrast (copy archived at swh:1:rev:fe8d51dc732f-dc0336532ecc47cb7b18c01dc8cf, *De and Horwitz, 2021b*).

## General

All protocols conformed to the guidelines provided by the US National Institutes of Health and the University of Washington Animal Care and Use Committee. Data were collected from two adult male rhesus macaques (*Macaca mulatta*). Each monkey was surgically implanted with a titanium headpost and a recording chamber (Crist Instruments) over area V1. Eye position was monitored continuously using either an implanted monocular scleral search coil or a digital eye-tracking system (SMI iView X Hi-Speed Primate, SensoMotoric Instruments).

## Task

The monkeys sat in a primate chair 1 m from a cathode ray tube (CRT) monitor (Dell Trinitron Ultrascan P991) in a dark room during the experiments. In a subset of sessions, the distance was reduced to 0.7 m and the pixel size was changed accordingly to preserve angular subtense. During white noise presentation, the monkeys fixated a centrally located dot measuring 0.2°×0.2° and maintained their gaze within a 1.6°×1.6° fixation window. During the closed-loop isoresponse measurements, the monkeys maintained their gaze within a 0.8°×0.8° window. Successful fixation was rewarded with apple juice, and fixation breaks aborted trials.

## Monitor calibration

Monitor calibration routines were adapted from those included in Matlab Psychophysics toolbox (*Brainard, 1997*; *Kleiner et al., 2007*; *Pelli, 1997*). The emission spectrum and voltage-intensity relationship of each monitor phosphor were measured with a spectroradiometer (PR650, PhotoResearch Inc). Stimuli were gamma-corrected in software to compensate for the nonlinearity of voltage-intensity relationships. The color resolution of each channel was increased from 8 to 14 bits using a Bits++ video signal processor (Cambridge Research Systems, Ltd). The monitor refreshed at 75 Hz and background was uniform gray (x=0.3, y=0.3, Y=55–75 cd/m$^2$). Tabulated primary emission spectra and gamma-corrected background RGB levels are available at https://github.com/horwitzlab/Chromatic_spatial_contrast (*De and Horwitz, 2021b*).

## Electrophysiological recordings

We recorded from well-isolated V1 neurons (RF eccentricity: 1.3°–5.9°, median=3.6°) using extracellular tungsten microelectrodes (Frederick Haer, Inc) lowered through the dura mater via hydraulic microdrive (Stoelting Co). Electrical signals were amplified, digitized at 40 kHz, and recorded using Sort Client software (Plexon, Inc). Action potentials were identified offline using Offline Sorter software.

## Experimental protocol

Each experiment consisted of three phases. During the first phase, spatiochromatic tuning was probed with a white noise pixel stimulus and data were analyzed online by spike-triggered averaging. During the second phase, the white noise stimulus was customized to the RF of each neuron. During the third phase, high-contrast images with the same spatial structure as the *Phase 2* stimulus were presented for relatively long durations (300 ms). Each of these phases is detailed below.

### Phase 1: pixel white noise

Each pixel white noise frame consisted of a 10×10 grid of pixels each of which subtended 0.2°×0.2° (*Figure 2A*; *Horwitz et al., 2007*). The stimulus changed every 13.33 ms. The intensity of each phosphor at each pixel was modulated independently according to a truncated Gaussian distribution with a standard deviation that was 15% of the physically achievable range. The space-time averaged intensity of each phosphor was equal to its contribution to the background.

Neuronal responses to the pixel white noise stimulus were analyzed by spike-triggered averaging (*Figure 2A*). In this analysis, the 15 frames preceding each spike were collected and averaged across

spikes. From these 15 STA frames, we selected online the one that differed most from the background on the basis of the sum of squared elements (p<0.0001, $\chi^2$ test) and identified pixels that differed significantly from the background (p<0.05, z-tests performed on each phosphor separately). These data were used to customize the white noise stimulus to the RF in *Phase 2* of the experimental protocol (see below). We selected for additional study only cells whose RFs consisted of at least two subfields with distinct chromatic preferences (*Figure 2A*).

The pixel white noise stimulus modulated neurons weakly for three reasons. First, individual stimulus pixels were small relative to V1 RFs. This was necessary to distinguish one RF subfield from another but resulted in each subfield being stimulated by independent pixel modulations that tended to cancel. Second, the pixels modulated rapidly, so multiple frames were effectively averaged together in the early visual system, prior to V1. Longer frame durations might have been preferable because spatial, not temporal, aspects of the response were of primary interest. Third, phosphor intensities were drawn from Gaussian distributions. Most of the probability mass of a Gaussian distribution is near the mean, which was identical to the background, so high-contrast pixels were improbable (*Figure 2— figure supplement 1A, B*). Non-Gaussian distributions complicate spike-triggered covariance analysis (*Paninski, 2003*).

## Phase 2: hyperpixel white noise

For each neuron with an STA containing at least two spatially distinct subregions (with distinct chromatic preferences), we created a custom 'hyperpixel' white noise stimulus by yoking the pixels within each of the two subfields (*Figure 2B*). Phosphor intensities at the two yoked collections of pixels (the two hyperpixels) were modulated according to the same Gaussian distributions used in *Phase 1*. Pixels outside of the RF were not modulated.

Cone weights were calculated from *Phase 2* of the experimental protocol. For each cell, we identified the STA frame that differed maximally from the background and computed a weighted average of this frame and the two flanking frames. The weight of each frame was proportional to the sum of squared red, green, and blue intensities relative to the background. We converted these weighted STAs to cone weights that are assumed to act on cone contrast signals (*Weller and Horwitz, 2018*). This procedure provided a pair of cone weights for each neuron, one set for each hyperpixel.

To compute a single set of cone weights for each neuron, we decomposed the weighted STA into a color weighting function and a spatial weighting function, defined as the first left- and right-singular vectors from a singular value decomposition (*De and Horwitz, 2021b*; *Horwitz and Albright, 2005*). This decomposition has the feature that the outer product of the color weighting function and the spatial weighting function (multiplied by the first singular value) is the rank-1 matrix that best approximates the weighted STA (*Eckart and Young, 1936*). Taken together, the color and spatial weighting functions captured 96.7±5.0% (mean ± SD) of the variance in the weighted STAs. The color weighting function was converted to cone weights as described above. Every spatial weighting function consisted of one positive and one negative weight, because only neurons with spatially opponent RFs were analyzed.

To examine how signals were combined across the two targeted subfields, we computed the 15-frame hyperpixel STA and convolved it with the reconstructed hyperpixel stimulus movie. This operation provides two numbers for each frame that represent how strongly each short, overlapping segment of the stimulus movie drove the two RF subfields. We visualized a firing rate map from these pairs of values by binning them and then computing the proportion of stimuli within each bin that preceded a spike (*Chichilnisky, 2001*). We also fit these data with linear and nonlinear models to examine integration across RF subfields quantitatively (see Hyperpixel white noise analysis of signal combination across subfields).

## Phase 3: isoresponse measurement

We selected the hyperpixel STA frame that differed most from the background and separated it into its two components, each of which stimulated one RF subfield with its preferred light (represented along the 45° and 135° directions in *Figure 3A*). We then linearly combined these two images in different proportions to create a family of stimuli that can be represented in the same plane used to construct the firing rate map in *Phase 2* (*Figure 2C*). The origin of the coordinate system represents

the gray background of the display. Direction represents the contrast between the two halves of the stimulus, and distance from the origin represents stimulus contrast relative to the background.

Contrast was quantified as follows. Each half of the hyperpixel STA was represented as a three-element unit vector. For example, if the STA indicated maximal sensitivity to increments of the red primary, half as much sensitivity to decrements of the green primary, and zero sensitivity to the blue primary, this vector would be [0.89 –0.46 0]. If the background was [0.5 0.5 0.5] (where '0' means completely off and '1' means maximum intensity), then possible stimuli included [0.589 0.454 0.5] (positive contrast), [0.411 0.546 0.5] (negative contrast), and [0.5 0.5 0.5] (zero contrast). To reduce RGB triplets to single numbers, we subtracted the background and then projected onto the unit vector: (e.g., [0.589 0.454 0.5]-[0.5 0.5 0.5]*[0.89 –0.46 0]$^T$=0.1). This calculation provides one of the two coordinates needed to represent the stimulus as a point on a plane. The other coordinate was calculated identically, using the other half of the STA. Note that this stimulus space is different for each neuron because the axes are derived from each neuron's hyperpixel STA.

## Contrast staircase procedure

To examine interactions between subfields, we identified collections of stimuli described above that each evoked the same number of spikes using the following procedure. On each trial, the computer presented a stimulus and counted spikes from the response latency, defined as the peak frame of the hyperpixel STA from *Phase 2*, until the end of the stimulus presentation. This spike count was compared to an experimenter-defined target response (*Figure 3—figure supplement 1A*). If the spike count was lower than the target response, the contrast of the image was increased by a factor of 1.35. If the spike count exceeded the target response, the contrast was decreased by a factor of 0.65. This process continued until a reversal occurred. A reversal is a response that exceeded the target response after having fallen below it on the previous stimulus presentation or a response that fell below the target having exceeded it on the previous stimulus presentation. After each reversal, the change in contrast per trial decreased by 25% (*Figure 3—figure supplement 1B*). The staircase halted after seven reversals or whenever the contrast exceeded the physical limitations of the display. Staircase termination points were taken as estimates of the contrast that evoked the target response. Presentations of stimuli in pairs of directions in the stimulus space were randomly interleaved to mitigate non-stationarity due to adaptation. Each stimulus was presented for 300 ms and was separated from the preceding and subsequent stimuli by more than 1 s.

## Cell screening

We recorded from 232 well-isolated V1 neurons and made isoresponse measurements from 98 of them. These 98 neurons were selected on the basis that they were well-isolated throughout all the three phases of the experimental protocol and had STAs with clear spatial structure. Neurons were classified as 'simple', 'double-opponent,' or 'other spatially opponent' on the basis of responses to hyperpixel white noise as described below.

## Spike-triggered covariance analysis

We distinguished simple cells from DO cells on the basis of cone weights, and cone weights are interpretable only under a linear model of signal combination across cone types (*Weller and Horwitz, 2018*). To identify neurons that are poorly described by a linear model, we computed the first principal component (PC1) of the spike-triggering stimuli orthogonal to the hyperpixel STA (*Horwitz et al., 2005*; *Rust et al., 2005*; *Touryan et al., 2002*). A PC1 that is larger than expected by chance reveals a nonlinear component of the cell's response to the white noise stimulus that cannot be captured by an output nonlinearity. We assessed the significance of the PC1 by randomly shifting spike trains in time relative to the *Phase 2* stimulus movie, recalculating the PC1, and obtaining its eigenvalue (*Rust et al., 2005*). This procedure was repeated 1000 times. If the largest eigenvalue from the unrandomized data exceeded 95% of the largest eigenvalues from the randomized data sets, we concluded that the PC1 was significant at the 0.05 level. Thirty-five neurons with a significant PC1 were classified as OSO on this basis.

Neurons lacking a significant PC1 were classified as simple if their L- and M-cone weights had the same sign, accounted for 80% of the total cone weight, and individually accounted for at least 10%. None of the simple cells we studied showed evidence of opponent input from the S-cones. Twenty-six

cells in our data set were categorized as simple. A cell was classified as DO if it lacked a significant PC1 and had a pair of cone weights of opposite sign. Cone weights of small absolute value were ignored; to be classified as DO, a neuron either had to have (1) an S-cone weight that accounted for at least 20% of the total in addition to a pair of opponent weights, or (2) L- and M-cone weights that accounted for at least 80% jointly and 20% individually. Using these criteria, we classified 1 S−M+L, 3 S+M−L, and 2 S−M−L cells as DO by the first criterion and 19 L−M cells as DO by the second criterion. Twenty-five cells were categorized as DO. The 12 neurons that did not meet the cone weight criteria for the DO or simple cell categories were classified as OSO, leading to a total of 47 in this category.

## Luminance tuning index

To test the robustness of the results to the cone weight classification criteria, we segregated simple cells from DO cells using an alternative approach. For each neuron, we calculated a luminance tuning index (*Figure 3—figure supplement 5*) by projecting the normalized cone weights of each cell onto a set of cone weights derived from the photopic luminous efficiency function. The luminance cone weights were estimated by regressing the *Vos, 1978* 2° photopic luminosity function onto the Stockman-Macleod-Johnson 2° cone fundamentals to find the best-fitting coefficients (0.83L+0.55M+0.03S) (*Stockman et al., 1993*; *Vos, 1978*). The luminance tuning index ranged from 0 to 1. Cells were classified as DO if their index value was <0.33 and if they lacked a significant PC1 (n=22). Cells were classified as simple if their index value was >0.67 and if they lacked a significant PC1 (n=27). The remaining 49 neurons were classified as OSO.

## Hyperpixel white noise analysis of signal combination across subfields

We fit the data from *Phase 2* of the experimental protocol with a GLM and a GQM.

The GLM had the form:

$$predresp = \frac{1}{1+e^{-(w_1 P_1 + w_2 P_2 + c)}}$$ (1)

where *predresp* is the predicted response of the neuron and $P_1$ and $P_2$ are the projection magnitudes of the 15-frame-long segments of the stimulus movie onto the two halves of the temporo-chromatic hyperpixel STA. $w_1$, $w_2$, and $c$ were fit using the Matlab routine *fitglm* to maximize the binomial likelihood of a spike. *Predresp* is constrained to be ≤1 because it represents a spiking probability, and the constant term, $e^{-c}$, captures the overall responsivity of the cell.

The GQM had the form:

$$predresp = \frac{1}{1+e^{-\left(w_1 P_1^2 + w_2 P_2^2 + w_3 P_1 P_2 + w_4 P_1 + w_5 P_2 + c\right)}}$$ (2)

## Evaluating the performance of generalized linear and quadratic models

We quantified the ability of the fitted models to predict whether or not each stimulus segment evoked a spike using ROC analysis (*Green and Swets, 1966*). Classification error was defined as one minus the area under the ROC curve (*Figure 2—figure supplement 2B*). To avoid overfitting, the model was fit with 90% of the data and tested on the remaining 10%. The white noise nonlinearity index (white noise NLI) for each cell was defined as:

$$Whitenoise\ NLI = log_{10}\left(median\left(\frac{Classification\ error_{GLM}}{Classification\ error_{GQM}}\right)\right)$$ (3)

where the median is taken is taken across 10 cross-validation data partitions.

## Error estimates of NLIs

The reliability of the NLIs was assessed using a jackknife resampling procedure. As described above, a GLM and GQM classification error was computed from multiple data partitions. An NLI was calculated from each pair of classification errors, and the spread of NLIs was quantified using the standard jackknife formula (*Efron and Tibshirani, 1994*).

## Model fits to isoresponse staircase termination points

To assess the linearity of signal integration across the gamut of our video display, we fit the staircase termination points from *Phase 3* with linear and quadratic models. Fitting was performed using a standard inbuilt Matlab routine for function minimization (*fmincon*) to minimize the Tukey-bisquare objective function (*Fox, 2002*).

Searches for stimuli that produced the target response were conducted in multiple directions of the stimulus space (e.g., *Figure 3A–C*), but angles were fixed. We therefore fit the data with a model that assumes radial error. The linear model can be written as:

$$A \, r \, cos\theta + B \, r \, sin\theta = 1 \tag{4}$$

$$r = \sqrt{x^2 + y^2}, \theta = tan^{-1}\frac{y}{x} \tag{5}$$

Where $x$ represents the projection of each image onto one hyperpixel of the STA and $y$ represents the projection onto the other hyperpixel (see *Phase 3: Isoresponse measurement*). $A$ and $B$ are fitted parameters. Fitting was performed on $log(r)$, not $r$, because contrast was adjusted multiplicatively, not additively, during the staircase procedure.

The quadratic model can be written as:

$$A \left( r \, cos\theta \right)^2 + B \left( r \, sin\theta \right)^2 + Cr^2 cos\theta sin\theta + Drcos\theta + Ersin\theta = 1 \tag{6}$$

where $A$, $B$, $C$, $D$, and $E$ are fitted parameters.

## Evaluating model fits to staircase termination points

We evaluated the quality of model fits by calculating the sum of Tukey-bisquared errors between the data and the model predictions. To avoid overfitting, we used leave-one-out cross-validation. The isoresponse nonlinearity index (isoresponse NLI) was defined as the median of the ratio of cross-validated linear model errors and quadratic model errors in logarithmic units.

$$Isoresponse \, NLI = log_{10} \left( median \left( \frac{Crossvalidated \, error_{Linearmodel}}{Crossvalidated \, error_{Quadraticmodel}} \right) \right) \tag{7}$$

NLI reliability was quantitated with the jackknife procedure described above.

## Hyperpixel white noise analysis of signal combination within subfields

To test the nonlinear model in *Figure 1C*, we fit the data collected during *Phase 2* of the experiment with GLMs and GQMs that predicted spikes on the basis of RGB values at individual hyperpixels. The models classified the RGB triplet on each frame as belonging to the spike-triggering or non-spike-triggering ensemble (*Figure 2—figure supplement 4*).

The latency of the response, which was needed to associate stimulus frames with spikes, was based on the time of the hyperpixel STA peak. Unlike the 'across-subfield' analysis, we only selected a single frame for 'within-subfield' analysis. Basing the within-subfield analysis on individual frames bypassed the assumption of linear temporal integration required to reduce multi-frame segments to single average RGB values and allowed us to focus on the integration of RGB values within a single subfield.

The GLM had the form:

$$predresp = \frac{1}{1+e^{-\left(w_1 R + w_2 G + w_3 B + c\right)}} \tag{8}$$

where $w_1$, $w_2$, $w_3$, and $c$ are fitted parameters. The output of the model is the probability of spiking given a triplet of background-subtracted red, green, and blue phosphor intensities ($R$, $G$, and $B$).

The GQM had the form:

$$predresp = \frac{1}{1+e^{-\left(w_1 R^2 + w_2 G^2 + w_3 B^2 + w_4 RG + w_5 GB + w_6 RB + w_7 R + w_8 G + w_9 B + c\right)}} \tag{9}$$

We used tenfold cross-validation to avoid overfitting and quantified the ability of the models to predict whether a stimulus frame preceded a spike or not using an ROC as described above. In total, 40 models were fit per neuron (one GLM and one GQM at each hyperpixel with tenfold cross-validation for each). To report a single value for each neuron, we averaged classification error rates across subfields and computed the median across cross-validation folds using the formula:

$$Within\ subfield\ NLI = log_{10}\left(median\left(\frac{\sum_{subfield=1}^{2} Classification\ error_{GLM}}{\sum_{subfield=1}^{2} Classification\ error_{GQM}}\right)\right) \quad (10)$$

### Drifting gratings

Two neurons were stimulated with drifting, sinusoidal gratings (two cycles per degree, 3 Hz, 1° diameter circular aperture) that modulated L- and M-cones with identical contrasts either in phase (L+M) or in anti-phase (L−M) (*Figure 7*). These neurons were not tested with the standard protocol.

## Acknowledgements

The authors thank Yasmine El-Shamayleh, Fred Rieke, Greg Field, and Jacob Yates for comments on the manuscript. This work was funded by NIH EY018849 to Gregory D Horwitz, NIH/ORIP Grant P51OD010425, and NEI Center Core Grant for Vision Research P30 EY01730 to the University of Washington and R90 DA033461 (Training Program in Neural Computation and Engineering) to Abhishek De.

## Additional information

### Funding

| Funder | Grant reference number | Author |
| --- | --- | --- |
| National Eye Institute | EY018849 | Gregory D Horwitz |
| Office of the Director | OD010425 | Gregory D Horwitz |
| National Eye Institute | EY01730 | Gregory D Horwitz |
| National Institute on Drug Abuse | DA033461 | Abhishek De |

The funders had no role in study design, data collection and interpretation, or the decision to submit the work for publication.

### Author contributions

Abhishek De, Formal analysis, Investigation, Methodology, Software, Visualization, Writing – original draft, Writing – review and editing; Gregory D Horwitz, Conceptualization, Methodology, Writing – original draft, Writing – review and editing, Funding acquisition, Supervision

### Author ORCIDs

Abhishek De http://orcid.org/0000-0002-2978-473X
Gregory D Horwitz http://orcid.org/0000-0001-5130-5259

### Ethics

This study was performed in strict accordance with the recommendations in the Guide for the Care and Use of Laboratory Animals of the National Institutes of Health. All of the animals were handled according to approved institutional animal care and use committee (IACUC) protocol (#4167-01) of the University of Washington. All surgery was performed under sevoflurane anesthesia, and every effort was made to minimize suffering.

### Decision letter and Author response

Decision letter https://doi.org/10.7554/eLife.68133.sa1
Author response https://doi.org/10.7554/eLife.68133.sa2

## Additional files

### Supplementary files

• Transparent reporting form

## Data availability

All data associated with this study are available at https://github.com/horwitzlab/Chromatic_spatial_contrast (copy archived at swh:1:rev:fe8d51dc732fdc0336532ecc47cb7b18c01dc8cf).

The following dataset was generated:

| Author(s) | Year | Dataset title | Dataset URL | Database and Identifier |
| --- | --- | --- | --- | --- |
| De and Horwitz | 2021 | Chromatic spatial contrast | https://github.com/horwitzlab/Chromatic_spatial_contrast | Github, fe8d51d |

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
