## [Editor Report]

There was a strong agreement that the work fills an important gap in the understanding of spatial integration of color signals in primate visual cortex and that the use of novel stimulus paradigms to study color responses in alert macaque provided important new information. There was also strong support for the analytical approaches used to categorize simple and double opponent neurons as well as appreciation for the quality of the discussion of the results.

---

## [Decision Letter]

**Decision letter after peer review:**

Thank you for submitting your article "Coding of chromatic spatial contrast by macaque V1 neurons" for consideration by *eLife*. Your article has been reviewed by 2 peer reviewers, and the evaluation has been overseen by a Reviewing Editor and Tirin Moore as the Senior Editor. The following individuals involved in review of your submission have agreed to reveal their identity: Bevil R Conway (Reviewer #1); David Brainard (Reviewer #2).

Your manuscript was well received by the reviewers. They felt that your work fills a major gap in the understanding of cortical processing of color signals and is likely to be of broad interest. They also felt that the experiments are well executed and paper well-written. However, they raised a number of concerns which need to be addressed for the work to be publishable in *eLife*. Below are the topics that will require substantial revision.

1. The extent to which the results are affected by the cell classification criteria. The reviewers raise several related points and provide some possible ways of addressing the issue (which will potentially require further analysis).

2. The implications of the results for models, including the issue of how stimuli were defined and the implications of the stimulus space that was used. It will be essential that enough information is provided in the paper that others could convert the display RGB data into device-independent stimulus spaces such as cone-contrast space.

3. The color properties of the subfields.

4. The impact of eye movements on the results.

5. Potential implications of your choice of stimulus space

6. Provide information about your display's RGB data that would allow their conversion into a cone contrast space

In addition to the main concerns listed above, the two reviews contain detailed comments and suggestions which you are likely to find useful, as you revise the manuscript.

*Reviewer #1 (Recommendations for the authors):*

Many studies of the chromatic spatial response properties of V1 cells have used heterochromatic gratings where the colors of the grating are fixed by some assumed theory (typically cone opponency through, e.g., DKL color space). I am aware of only three studies that have addressed how V1 cells sum responses across receptive field subunits: one showed responses for only two cells, one clearly linear and one clearly non-linear (Conway, 2001); a second study focused on the integration of color signals in time, rather than in space (Conway et al., 2002); and a third measured responses to pairs of cone-isolating spots and found that L-M cells were linear while S-dominated cells were often non-linear (Conway and Livingstone, 2006). The approach in the present report, combining the white-noise analysis and the "hyperpixel" contrast approach are very powerful (and clever), overcoming the limited set of colors used in the previous reports and adding substantially more quantitative resolution. But the analysis of the results do not seem to fully explore the richness of the data, and leave us with an unsurprising conclusion that confirms what we already know-double-opponent cells include a population of linear neurons. As the authors point out, previously published spatial mapping studies of cone-opponent neurons have recovered spatial structure, and the technique used in those studies only reveal spatial structure if the neurons show some degree of linearity. It is well established that the majority of V1 cells in primates are non-linear (complex) and the minority of simple cells in primate are linear (by definition). So it is not surprising that double-opponent cells in the present work are more likely to be linear compared to the somewhat random sample of non-DO/non-simple cells, which were, by definition, classified as nonlinear (discussion, last paragraph, pg. 14).

The value of the present study would seem to rest on the distinction between simple and DO cells and the selection criteria for the NSNDO population. I appreciate that there is some arbitrariness in how cells are categorized, and the authors have attempted to show that the main conclusion is maintained if category assignments are made using two different criteria. But I wonder if a simpler approach, which I think cleaves nature more naturally along the joints, would be to define cells based on significance of S cone input, and then on whether the L and M inputs have the same sign or different sign. This approach would leave the categorization of "simple" and "double opponent" to the discussion, and it would preclude an objection of the results/conclusions on the grounds of miss-categorization. The anatomy suggests that neurons carrying S cone signals are distinct, and probably have different patterns of wiring; and cells that respond with the same sign to L increments and M increments (without responding to S cone stimuli) would seem to be functionally different from neurons that respond in an L-vs-M opponent fashion. Moreover, there is considerable data (from at least two groups, Horwitz and Conway), that V1 cells with strong S-cone input show striking nonlinearities. (Are the greater number of NLI cells in the DO population and the simple cell population those with stronger S cone responses?) The physiological responses documented in the present report would be more valuable, in my opinion, if they provided clues that link with the anatomy, rather than being constrained by categorization based on potentially arbitrary physiological criteria (such as "simple" vs. "double opponent"). Of course, this is only a suggestion, and the authors should be free to analyze and present their data as they see best.

The analysis side-steps the important temporal dynamics of the cell responses. As with simple cells and double-opponent cells, responses are often push-pull in space and in time. It would seem important to test the extent to which the assigned linearity score of a neuron in space is predictive of its linearity score in time.

The hyperpixel paradigm involves presenting the stimuli at a fixed spatial location while the monkeys maintain fixation. The validity of the approach depends on the extent to which a given receptive field subregion is stimulated reliability and exclusively despite small eye movements. Some assurance (ideally with data) that this is the case would be helpful. My worry is that the interpretability of the results (given the eye movements) depends on the neurons being linear.

Finally, the paper does not explore the color tuning of the subunits, except to state that "the colors on the two sides of the hyperpixel STA were complementary or nearly so". The Pearson's r analysis is insufficient: it tells us nothing about the color tuning, and high correlation values can be obtained for trivial reasons. The paradigm illustrated in Figure 2B provides a terrific chance to determine, directly, what the color tuning is of the separate subunits, and the extent to which the color tuning of one subunit of a given cell can predict the color tuning of the other subunits. This is important because color tuning inferred from cone weights suggests a bias (M+S vs L) that the paper is poised to directly test. Such an analysis could also help tease apart the distinction between simple and DO cells.

Other comments

Methods. The procedure by which different proportions of the two images were combined to create the family of stimuli is not entirely clear to me. The proportionality needs to be done by quantifying the stimuli in some way. The authors state that this was done by defining the 45 degree direction as the projection of the RGB values onto one half of the STA. But here's where I get tripped up: the goal of defining the family of stimuli is to then assess whether the neuron responses map on a line in the space or on a curve. If the space is arbitrarily defined, then a given neural response can be captured either by a line or a space depending on how the space is defined. I need some way out of this (apparent) circularity.

Intro. Given the relevance to simple cells, it might be useful to have a reference to the experimental evidence of simple cells in NHPs (there was a time, not too long ago, when dogma was that NHPs did not have simple cells).

Results. Given the validity of the results hinges on the distinction between simple, double-opponent, and NSNDO cells, I think it would be helpful to present the categorization criteria in the Results section, rather than relegating this information to the methods.

Results, pg. 6. The "hyperpixel" approach is analogous to the two-bar approach used in earlier studies. Except that in the earlier study the pair of bars were presented across the RF, which allowed quantification of more than two subunits. In the present work, did the stimulus always comprise just two halves? What if the cell had more than two subunits, as is not uncommon for simple and DO cells (and appears to be true for the example cell in Figure 2A)?

Results, pg. 6. I was a bit tripped up by the flow of the results: the STA analysis (2A) is what is used to recover the NLI histograms (2D-2F); the panels in 2B and 2C show a separate experiment. Perhaps regrouping the figures accordingly would help avoid the confusion that Figure 2D-2F quantify what is shown in Figure 2C (which is what would be logically assumed given the current sequence of panels and how they are described in the text).

Results, pg. 7. Is it not true that the extra parameters of the nonlinear model will also make it more likely that cross-validated data will be better fit by it? In other words, using cross validation doesn't get over the unfairness of comparing the GQM and GLM models, does it?

Results, pg. 8. The Kruskal-Wallis test outcome is driven entirely by the NSNDO population, which likely includes complex cells. It is well established that complex cells and simple cells differ in linearity, so it's not clear to me what this NLI analysis recovers. As the authors point out, the reverse-correlation mapping studies published previously only recover spatial structure if the cells have a linear component. So we already know that DO cells are more linear than the average V1 cell.

Results, pg. 10. The discussion of the necessity of the technique is helpful. Is this is the same logic underlying the use of responses to L-decrements to measure suppression by L (for green-ON cells) and M-decrements to measure suppression by M (for red-ON cells) by others?

Results, pg. 10. It would be helpful to state what conclusion (curved or linear) the authors interpret Figure 2C as showing. I'm assuming curved because the pattern of gray shading forms an arc around the 0,0 point. But then what are the dashed white lines? If they are intended to show model fits of stimuli that elicit the same firing rate, then I think something is wrong with that algorithm because the data do not, to my eye, align with the dashed lines. If they are intended to show theoretically which stimuli would have the same firing rate if the neuron were linear, then I'm confused why the lines are tilted.

Results, pg. 10. It seems odd to set the target firing rate to be lower than significantly (substantially) above baseline firing rate for any cells (even three). It would be useful to know the range, mean, and confidence limits around the mean, for the target firing rates (in terms of % above background firing rate).

Discussion, pg. 12. I appreciate the point the authors are trying to make, but as framed here, it seems weak ("we used a visual stimulation paradigm that only works if the cells are linear to test if the cells are linear").

Discussion, pg. 13. The prior study discussed by the authors concluded that the responses to pairs of bars was linear. Another earlier study (Conway, 2001) showed two-bar interaction maps for two cells, one of which was clearly non-linear (and the non-linear cell was the one with strong S-cone input).

Discussion, pg. 13. The authors state that the prior work made an assumption that excitatory responses to an increment were the same size but opposite sign to suppression of a contrast decrement. It's not clear to me that this was an assumption rather than a test. The reverse-correlation procedure used in the earlier work used as a baseline the responses to stimuli presented well outside the receptive field, which allows the assumption to be directly tested.

Discussion, pg. 16 "In either case, DO cells are a likely basis for the chromatic sensitivity of color-sensitive complex cells (Michael, 1978)." This isn't clear to me. It seems equally plausible (arguably more plausible) that color-sensitive complex cells could derive color tuning by direct sampling of geniculate inputs.

Figure 3. It would be helpful to show the STA for each of the example cells. The distribution of data points in Figure 3A is stated in the results as being linear, but it looks curved to me (the two outliers at the far left notwithstanding). None of the example cells in Figure 3 are L-vs.-M cells, which is odd since the majority of "classical" double-opponent cells are defined by L-vs.-M opponency. By contrast, all the cells in Figure 3A could plausibly be simple cells. I'm a bit confused how meaningful the responses of a cell are if they fall on a line that goes through the origin. Responses at the origin are to a null stimulus (blank gray), which indicates that the target firing rate is nota much different from baseline.

*Reviewer #2 (Recommendations for the authors):*

General. I think this paper would benefit from a sharpening of terminology and usage throughout. For example, always being explicit about what sort of non-linearity is being considered in any given sentence, and about which of the two types of white noise experiments is being referred to in each place where a reference is made. I could almost always figure out it what was intended from context, but that was extra work and left some uncertainty, as well distracted from following the flow of ideas. Specific examples given in some of the comments below.

Starting on page 3 and throughout. A key to understanding this paper is to understand the taxonomy the authors have applied to the neurons they recorded from. I struggled with the simple cell terminology, though. The criteria the authors use are different from more conventional ways used distinguish simple and complex cells. I think that if one applied F1/F0 criteria using luminance grating stimulation, one might well call at least some of the DO cells simple, as well as possibly some of the cells not studied. The criteria here are that the hyperpixel white noise analysis shows that cone contrast inputs are combined linearly and that the RFs are not cone opponent. I would have an easier time if the term simple cell were not used, to avoid confusion with the conventional taxonomy. Perhaps single opponent cells, or spatially-opponent non-cone-opponent cells. Thesse suggestion don't capture the linearity aspect of the criteria, but nor does the term double opponent and the authors are content with that. Maybe both terms should more clearly denote the linearity criteria. With a different choice of terminology, the relation to the conventional simple/complex taxonomy could be handled entirely in the discussion, with some care devoted to treating possible differences in classification. Avoiding the simple cell terminology might also allow ease exposition on the possible relation between NSNDO cells in the current study and chromatically sensitive complex cells, where I think more elaboration would be helpful.

Page 3. Introduction: "The first stage of the primate visual system at which spatial comparison of spectral signals is implemented by individual neurons is area V1." – This statement is mysterious to me. Spatial comparisons are made at the first synapse, as signals from cones interact with signals from horizontal cells, and these interactions can involve spectral differences as signals from different classes of cones participate. Perhaps the authors mean something very specific when the qualify with "by individual neurons" but it is not clear what this is.

Page 4. I think more should be said about the 134 neurons that were recorded from and not studied further. These don't have a name in the current study and it is not entirely clear to me what caused them to be excluded. Perhaps only that they did not show any spatial opponency in the initial pixel white noise analysis (page 6), but I'm not sure that's the only reason so more detail on the exclusion criteria would be helpful.

Page 5. I agree with the authors that equating stimulus contrast across chromatic directions is a difficult and subtle matter, and that whenever possible it is useful to look for ways to frame questions that are robust to such equation. I diverge in that I don't see that their method avoids this difficulty. Just as one example, the frame selection criteria (pp. 21-22) are based on deciding which STA frame differs most from the background. That calculation is using some metric across color directions. Similarly with the criterion used to decide that the chromatic preferences of the two subfields of a DO neuron differ from each other.

Page 12. While we're on this point, the authors have used device primary RGB as their metric. The justify the reasonableness of this choice by writing "The fact that DO cells and simple cells combined light intensities across space approximately linearly in this representation shows that it is a reasonable one for analyzing V1 neurons." This seems odd, first because it assumes the answer to the question being studied in this paper. Second, because if the combination is revealed to be linear in one choice of color representation, it will also be linear in another choice that is a linear transformation of the first. (Indeed, the authors themselves say this in the sentence after the one quoted.) So I don't see how the linearity of the result tells us much about whether the representational choice was good. If nothing else, I would ask that the authors provide enough information about their device so that others can transform their data into a device independent representation such as LMS contrast with respect to a well-defined choice of cone fundamentals.

Page 6. Figure 2B. I assumed the noise is independent across color channels in RGB, but would be good to say so explicitly.

Page 8. Figure 3. What should we make of the high indices for some DO cells, and the low indices for many NSNDO cells? Both distributions suggest at least that possibility that neither group is a homogeneous population with respect to non-linear spatial summation. More discussion of this seems worthwhile.

Page 10. How were the target firing rates chosen?

Page 13. "Nearly all quantitative studies of V1 RFs have used achromatic stimuli". Perhaps worth listing the studies of this sort that have used chromatic stimuli. Also, I think that by "V1 RFs" the authors mean "V1 spatial RF properties" and this might be worth spelling out.

Page 13. "This maximal response exceeded the response to either bar in isolation, consistent with linearity as well as with other models." Might make explicit that the linearity in this sentence is the linearity of spatial summation, as well as that the reason the result could be consistent with summation linearity is that there could be a response non-linearity after spatial summation.

Page 14. "Reclassifying cells …. " There's a lot of cell classification in this paper; perhaps sharpen here to emphasize that it is the simple/DO dichotomy classification being referred to here.

Page 14. "Most other recent studies of DO cells used cone-isolating stimuli, which cannot reveal interactions among cone types (Conway, 2001; Johnson et al., 2001, 2004; Conway and Livingstone, 2006; Johnson et al., 2008)." Johnson et al., used mixtures in some of their reported measurements.

Page 14. "We classified neurons with nonlinear responses to white noise as NSNDO. This criterion was necessary to satisfy the assumptions underlying the conversion of the STA to cone weights." (A) In the first sentence, "nonlinear" seems too general since the paper believes it worthwhile to check for a specific type of nonlinearity in the simple and DO cells. (B) I don't understand this because the authors go on to apply the conversion of the STA to cone weights with the NSNDO cells, in that such conversion is used for the test in Phase 2, as shown in Figure 3F. (C) Here is an other places, it would be helpful to say "hyperpixel white noise" rather than just "white noise"

Page 15. "Pooling LGN afferents with the same sign (ON or OFF) creates non-opponent spectral sensitivity." I'm not following what is meant here, need more detail on what is being assumed about chromatic properties of the afferents as well as exactly what is being pooled.

Page 17 and Figure 4. If the authors are going to show this data can they say anything about how the results for the experiments that are the focus of the current paper would come out for these two cells? That would connect these preliminary data better to the rest of the paper. And, did the authors find any cells like the one shown in 4C, but with the responses to luminance and chromatic gratings flipped?

Page 22. Please provide criteria used to decide whether the two subfields had distinct chromatic preferences, and how this related to cell classification. On page 7, correlations (computed how?) between the two halves of the STA are given, and it sounds like this is result obtained before selecting or classifying cells. One page 24 a percent variance explained number is given, again independent of cell classification. But I'm not entirely sure that this property wasn't required for cells to be included for further study.

Page 22. The authors note that using small pixels and a fast frame rate in the white noise experiments leads to the neurons being weakly driven. They might comment/think about whether the choices they made for these parameters are optimal – if they were to do the work again, could they consider using larger white noise pixels and a slower refresh rate to obtain better responses? That is, small/fast is at one end of a continuum, and maybe some other choice would be better for future work of this sort. Similarly, any requirement for use of the Gaussian distribution on the noise? Maybe a uniform or some other distribution would produce larger drive.

Page 22, last paragraph and into p. 23. The paragraph starts off talking about frames as six-dimensional vectors, but by the end it sounds like two separate 3-dimensional projections are done, yielding the two scalars one for each subfield. Would be clearer, I think, if only one of the two ways of talking about it is used here. Also might remind reader that this is the hyperpixel white noise STA, not the pixel white noise STA.

Page 23. We visualized a firing rate map from these projections by computing the ratio of spike-triggered stimuli to the total stimuli." I don't understand what metric of stimuli is being used in the ratio, which visualization in the paper this refers to, nor how the ratio of stimuli as described provides a map of firing rate.

Page 23. "Direction represents the overall contrast between …" Worth noting that what these axes represent in stimulus space is cell-specific, in that the axes for each cell are derived from that cell's hyperpixel STA. Also, although this is a matter of convention and up to the authors, I would find it more intuitive if the STA axes were the x and y axes of the plot, not the 45 and 135 directions.

Page 24. 232 neurons were recorded from. 98 are reported on here. What can we say about the rest, and why they were not included here. The one clear exclusion criterion is that some neurons had only a single spatial region revealed by the pixel white noise STA. Was that the only reason? On this point, I don't think NSNDO is the best term for the set of neurons out of the 98 that were not classified as simple or DO, because of all the other neurons that were not included and that also weren't classified as simple or DO.

Page 24. Is it a theorem that if you take the first row singular vector and the first column singular vector, that this is the optimal color-space separable accounting of the STA? Or could a joint analysis do better?

Page 24-25. On page 25, the classification of cells into simple, DO, and NSNDO is described. Because the term "white noise" is used to describe the protocol used for the classification and because this applies to both Phase 1 (pixel white noise) and Phase 2 (hyperpixel white noise) of the experiment, it would be good to remind the reader that it is (I think) the hyperpixel white noise that's relevant here. Indeed, I would try to be explicit each time in the paper the term white noise or STA is used, which of the two it refers to. Also, why is a substantial S cone input a requirement to be a DO cell? This seems like it would exclude cells with perfectly good spatially opponent L-M inputs. And, I'm not clear what happens to such cells, if there were any. I think from the next sentence that like neurons with a significant PC1, they get called NSNDO. But because the flow of this section had me thinking at the start that it was just those neurons with signifiant PC1 that were NSNDO, I'm also not clear on this. If they are NSNDO, then it is worth being explicit elsewhere that heterogeneous criteria were used to place neurons in this category, and perhaps looking at NLI separately for cells that arrived as NSNDO viat different paths.

Page 26. I'm surprised that Equation 1 doesn't include an amplitude parameter that captures the overall responsivity of the cell.

Figure 2 —figure supplement 1. I am not sure what is in panel D. Are these from Phase 2? And how does the 5th and 95th percentile information mentioned in the caption show up in the figure? More generally, I think the point of this figure is to provide a sense of how much harder the Phase 3 stimuli drive the cells than the Phase 1 and 2 stimuli, but I am not seeing this in the figure itself. Where are the Phase 3 data?

Figure 2 —figure supplement 2. Can you clarify the temporal parameters that go into this analysis?

Figure 3 —figure supplement 3. The staircase shown suggests that using the last stimulus presented does not give as good an estimate of stimulus that produces the criterion firing rate as would an analysis of the whole staircase. Do I understand that the last stimulus magnitude is what was used? If so, please elaborate why. If not, please clarify.

---

## [Author Response]

Reviewer #1 (Recommendations for the authors):Many studies of the chromatic spatial response properties of V1 cells have used heterochromatic gratings where the colors of the grating are fixed by some assumed theory (typically cone opponency through, e.g., DKL color space). I am aware of only three studies that have addressed how V1 cells sum responses across receptive field subunits: one showed responses for only two cells, one clearly linear and one clearly non-linear (Conway, 2001); a second study focused on the integration of color signals in time, rather than in space (Conway et al., 2002); and a third measured responses to pairs of cone-isolating spots and found that L-M cells were linear while S-dominated cells were often non-linear (Conway and Livingstone, 2006). The approach in the present report, combining the white-noise analysis and the "hyperpixel" contrast approach are very powerful (and clever), overcoming the limited set of colors used in the previous reports and adding substantially more quantitative resolution. But the analysis of the results do not seem to fully explore the richness of the data, and leave us with an unsurprising conclusion that confirms what we already know-double-opponent cells include a population of linear neurons. As the authors point out, previously published spatial mapping studies of cone-opponent neurons have recovered spatial structure, and the technique used in those studies only reveal spatial structure if the neurons show some degree of linearity. It is well established that the majority of V1 cells in primates are non-linear (complex) and the minority of simple cells in primate are linear (by definition). So it is not surprising that double-opponent cells in the present work are more likely to be linear compared to the somewhat random sample of non-DO/non-simple cells, which were, by definition, classified as nonlinear (discussion, last paragraph, pg. 14).The value of the present study would seem to rest on the distinction between simple and DO cells and the selection criteria for the NSNDO population. I appreciate that there is some arbitrariness in how cells are categorized, and the authors have attempted to show that the main conclusion is maintained if category assignments are made using two different criteria. But I wonder if a simpler approach, which I think cleaves nature more naturally along the joints, would be to define cells based on significance of S cone input, and then on whether the L and M inputs have the same sign or different sign. This approach would leave the categorization of "simple" and "double opponent" to the discussion, and it would preclude an objection of the results/conclusions on the grounds of miss-categorization. The anatomy suggests that neurons carrying S cone signals are distinct, and probably have different patterns of wiring; and cells that respond with the same sign to L increments and M increments (without responding to S cone stimuli) would seem to be functionally different from neurons that respond in an L-vs-M opponent fashion. Moreover, there is considerable data (from at least two groups, Horwitz and Conway), that V1 cells with strong S-cone input show striking nonlinearities. (Are the greater number of NLI cells in the DO population and the simple cell population those with stronger S cone responses?) The physiological responses documented in the present report would be more valuable, in my opinion, if they provided clues that link with the anatomy, rather than being constrained by categorization based on potentially arbitrary physiological criteria (such as "simple" vs. "double opponent"). Of course, this is only a suggestion, and the authors should be free to analyze and present their data as they see best.

Thank you for these suggestions. We have added a new analysis to the manuscript showing the relationship between the magnitude of S-cone weight and NLI. As the Reviewer predicted, neurons with large S-cone weights tend to be nonlinear. However, some neurons in our data set appeared to receive little S-cone input but nonetheless were highly non-linear. Defining cone weights for these neurons is difficult. Our estimates of how much input a neuron receives from the three cone types depends critically on how linearly these inputs are combined whereas our estimates of linearity are not trivially influenced by the strength of input from the three cone types. For these reasons, we have maintained the original criteria (screening first on the basis of linearity and second on the basis of cone weights) for defining the three groups of cells. However, we have changed the Introduction to more clearly explain these criteria.

The analysis side-steps the important temporal dynamics of the cell responses. As with simple cells and double-opponent cells, responses are often push-pull in space and in time. It would seem important to test the extent to which the assigned linearity score of a neuron in space is predictive of its linearity score in time.

We agree that the integration of visual information over time by V1 neurons is an important issue, but the data we collected are ill-suited for investigating this issue with the care it deserves. Ideally, to measure temporal linearity, we would have probed neurons with pairs of stimuli that are identical in space but separated in time (as in Conway and Livingstone, 2006). Instead, we probed neurons with pairs of stimuli that were separated in space but coincident in time, which is more appropriate for measuring spatial than temporal linearity.

Nevertheless, inspired by the Reviewer's suggestion, we analyzed our data for signs of push-pull excitation-inhibition, which manifests in OFF responses following the removal of a suppressive stimulus. For each neuron, we examined responses following the disappearance of a stimulus and measured the correlation of these responses with the isoresponse NLI. We would like to have have restricted this analysis to the removal of *suppressive* stimuli, but few of the stimuli presented were suppressive. As a proxy, we considered only trials in which no spikes were generated during the stimulus presentation period (consistent with the stimulus having been suppressive). We counted spikes from the disappearance of the stimulus (plus an additional latency derived from the hyperpixel STA) until 150 ms after stimulus disappearance. Fixation breaks during this epoch aborted trials. We found no significant relationship between the average OFF response and the isoresponse NLI (r=-0.01, p=0.89, Spearman’s rank correlation).

We also considered a stricter criterion, considering only neurons for which the average firing rate before the stimulus presentation (the baseline firing rate) was > 0 spikes per second (71/96 neurons satisfied the criteria). This allowed us to more easily identify suppressive stimuli. Even within this subset of neurons, we still did not find a significant correlation (r=0.01, p=0.84, Spearman’s rank correlation). The fact that we did not present many suppressive stimuli may have prevented us from observing strong OFF responses.

In interpreting these data, we note that linearity can be dissociated from OFFresponses; a linear neuron with a biphasic temporal impulse response produces an OFF response, but a linear neuron with a monophasic temporal impulse response does not (Author response image 1) .

**Author response image 1. sa2fig1:** Two hypothetical, purely linear L-M neurons stimulated with L-cone isolating contrast increments and decrements. If the temporal impulse response is monophasic (**A**), then the response to a step (**B**) is sustained, and no OFF-response occurs when the stimulus disappears. (**C** and **D**) same as **A** and **B** but for a neuron with a biphasic temporal impulse response. In this case, the disappearance of a suppressive stimulus produces an OFF response (dashed gray line).

The hyperpixel paradigm involves presenting the stimuli at a fixed spatial location while the monkeys maintain fixation. The validity of the approach depends on the extent to which a given receptive field subregion is stimulated reliability and exclusively despite small eye movements. Some assurance (ideally with data) that this is the case would be helpful. My worry is that the interpretability of the results (given the eye movements) depends on the neurons being linear.

We have included new analyses of fixational eye movements to the revised manuscript. Differences in fixation eye movements across neurons of different type were subtle, if present. This is expected: fixational eye movements shift the stimulus on the RF, but these shifts tend to be small with respect to the hyperpixels. Eye movements can make a two-subfield RF appear to have a single subfield (if the two hyperpixels spend an equal amount of time stimulating each subfield), but they cannot make a one-subfield RF present as an STA with multiple spectrally distinct subregions. All of the neurons described in this report had STAs with multiple spectrally distinct subregions. Moreover, we never observed a neuron that had two RF subregions in response to the pixel white noise but only one in response to hyperpixel white noise.

Importantly, in *Phase 3* of the experiment, images were flashed briefly, spikes were counted, and stimuli were not averaged together. The fact that we were able to cancel an increase in preferred color on one side of the RF with a similar change in light on the other side attests to the stability of the image on the retina (opponent regions of the RF were successfully targeted). We are unsure how eye movements might affect the interpretation of the data differentially for linear and nonlinear cells. Perhaps the reviewer is alluding to the fact that eye movements could cause a linear neuron to have a significant PC1 if these movements bring individual hyperpixels onto multiple subfields of different spectral sensitivity. Through this mechanism, eye movements would be expected to increase the significance of the PC1 of linear neurons. In this sense, our PC1 criterion is conservative.

Finally, the paper does not explore the color tuning of the subunits, except to state that "the colors on the two sides of the hyperpixel STA were complementary or nearly so". The Pearson's r analysis is insufficient: it tells us nothing about the color tuning, and high correlation values can be obtained for trivial reasons. The paradigm illustrated in Figure 2B provides a terrific chance to determine, directly, what the color tuning is of the separate subunits, and the extent to which the color tuning of one subunit of a given cell can predict the color tuning of the other subunits. This is important because color tuning inferred from cone weights suggests a bias (M+S vs L) that the paper is poised to directly test. Such an analysis could also help tease apart the distinction between simple and DO cells.

Thank you for this suggestion. We have added a new analysis of cone weights from the two hyperpixels separately. This analysis more clearly shows the distinction between simple cells and DO cells, documents the fact that the RF subfields of neurons in both groups have approximately opposite spectral tuning, and confirms the M+S vs L bias that has been documented before.

Other commentsMethods. The procedure by which different proportions of the two images were combined to create the family of stimuli is not entirely clear to me. The proportionality needs to be done by quantifying the stimuli in some way. The authors state that this was done by defining the 45 degree direction as the projection of the RGB values onto one half of the STA. But here's where I get tripped up: the goal of defining the family of stimuli is to then assess whether the neuron responses map on a line in the space or on a curve. If the space is arbitrarily defined, then a given neural response can be captured either by a line or a space depending on how the space is defined. I need some way out of this (apparent) circularity.

Thank you for pointing out this confusing part of the manuscript. The reviewer is correct that isoresponse contours have different shapes in different color spaces. However, lines in one color space remain lines in all color spaces that are related by a linear transformation. The color space we used is related to many conventional color spaces by a linear transformation (e.g. cone-contrast and DKL). This is clarified in the revised manuscript.

At each of the two hyperpixels, the STA reveals the neuron's relative sensitivity to the red, green, and blue primaries (and, by extension, the relative sensitivity to modulations of the L-, M-, and S-cone photoreceptors). We describe this sensitivity with a threeelement, unit vector. For example, if the STA indicated maximal sensitivity to increments of the red primary, half as much sensitivity to decrements of the green primary, and zero sensitivity to the blue primary, this vector would be [0.89 -0.46 0]. If the background was [0.5 0.5 0.5] (where "0" means completely off and "1" means maximum intensity), then possible stimuli at this location included [0.589 0.454 0.5] (reddish), [0.411 0.546 0.5] (greenish), and [0.5 0.5 0.5] (invisible). To express the contrast of each stimulus with a single number, we subtracted the background and then projected onto the unit vector: ([0.589 0.454 0.5]-[0.5 0.5 0.5])*[0.89 -0.46 0]' = 0.1. This calculation provides one of the two coordinates needed to represent the stimulus as a point on a plane. The other coordinate was calculated by performing the identical procedure on the other half of the STA. We have added the above explanation to the revised text.

Intro. Given the relevance to simple cells, it might be useful to have a reference to the experimental evidence of simple cells in NHPs (there was a time, not too long ago, when dogma was that NHPs did not have simple cells).

References to the existence of simple cells in NHPs have been added to the Introduction.

Results. Given the validity of the results hinges on the distinction between simple, double-opponent, and NSNDO cells, I think it would be helpful to present the categorization criteria in the Results section, rather than relegating this information to the methods.

We have expanded the explanation of the cell classification criteria in the Results section.

Results, pg. 6. The "hyperpixel" approach is analogous to the two-bar approach used in earlier studies. Except that in the earlier study the pair of bars were presented across the RF, which allowed quantification of more than two subunits. In the present work, did the stimulus always comprise just two halves? What if the cell had more than two subunits, as is not uncommon for simple and DO cells (and appears to be true for the example cell in Figure 2A)?

We did occasionally observe simple cells (and less frequently, DO cells) with > 2 subfields. For cells with more than 2 subfields, we selected for visual stimulation the most prominent pair of subfields (the three-subfield neuron in Figure 2A is a good example). We wholeheartedly agree that measuring spatial integration across multiple (> 2) subfields is important, but the three-phase, closed-loop technique implemented in the current study does not scale well beyond 2 subfields.

Results, pg. 6. I was a bit tripped up by the flow of the results: the STA analysis (2A) is what is used to recover the NLI histograms (2D-2F); the panels in 2B and 2C show a separate experiment. Perhaps regrouping the figures accordingly would help avoid the confusion that Figure 2D-2F quantify what is shown in Figure 2C (which is what would be logically assumed given the current sequence of panels and how they are described in the text).

Thank you for alerting us to this confusing issue. The NLI histograms (Figures 2D-2F) are computed from responses to the hyperpixel white noise (Figure 2B), and not the pixel white noise (Figure 2A). We have modified each mention of white noise or STA in the manuscript to specify whether we are referring to the pixel or the hyperpixel stimulus.

Results, pg. 7. Is it not true that the extra parameters of the nonlinear model will also make it more likely that cross-validated data will be better fit by it? In other words, using cross validation doesn't get over the unfairness of comparing the GQM and GLM models, does it?

Yes, we believe that cross-validation gets over the unfairness of comparing the GQM and GLM models. A non-cross-validated comparison would be unfair because the GQM makes the identical predictions to the GLM if some of the parameters of the GQM are set to zero. If these parameters are allowed to be something other than zero, the GQM will fit the data better unless the best-fitting values of these parameters happen to be zero, in which case there will be a tie.

When the data used to fit the model differ from the data used to test the model, as is the case in cross-validation, the GQM no longer enjoys this advantage over the GLM. The GQM will continue to fit the *training* data better than the GLM does, and this is for one of two reasons: The GQM may fit genuine structure in the data (patterns that are consistent across datasets produced by the same data generating mechanism) that the GLM does not fit. In this case, we expect the GQM to fit a new data set (or, equivalently, a held-out data set) better than the GLM does. Alternatively, the GQM may fit noise (patterns that are specific to the particular dataset being fit) that the GLM does not fit. In this case, we expect the GQM to fit a new data set worse than the GLM does.

Results, pg. 8. The Kruskal-Wallis test outcome is driven entirely by the NSNDO population, which likely includes complex cells. It is well established that complex cells and simple cells differ in linearity, so it's not clear to me what this NLI analysis recovers. As the authors point out, the reverse-correlation mapping studies published previously only recover spatial structure if the cells have a linear component. So we already know that DO cells are more linear than the average V1 cell.

Yes, the Kruskal-Wallis test outcome is driven by the OSO (previously called “NSNDO”) cells. However, a spatially structured STA was a requirement for inclusion in this study. Pure complex cells, which respond identically to contrast increments and decrements, were not studied. It seems likely that some neurons in the OSO category responded to contrast increments and decrements in the same RF locations, but with different sensitivity to each, and therefore had STAs with some spatial structure. We suspect that many of these cells would have been classified as complex cells by other procedures, but we cannot know for sure.

The white noise NLI is related to the spike-triggered covariance criterion that was used to exclude complex cells from the simple and DO cell categories but is not redundant with it. Please see our response to Reviewer 2, comment 20 for a more complete explanation of how these criteria differ.

Results, pg. 10. The discussion of the necessity of the technique is helpful. Is this is the same logic underlying the use of responses to L-decrements to measure suppression by L (for green-ON cells) and M-decrements to measure suppression by M (for red-ON cells) by others?

There may be a connection between the advantages we lay out for the isoresponse method and the use of responses to L-decrements to measure suppression by L and the use of M-decrements to measure suppression by M, but this connection is not obvious to us.

In this section, we focus on responses to pairs of stimuli and to each member of a pair separately. The key piece of logic is that two stimuli, when presented together, may drive a response that is not equal to the sum of the responses to the individual members of the pair but may still be predictable from these individual responses. The use of L-decrements to measure suppression by L and the use of M-decrements to measure suppression by M requires an assumption that two types of stimuli exert equal and opposite effects on the firing rate, which is not required in the above explanation.

Results, pg. 10. It would be helpful to state what conclusion (curved or linear) the authors interpret Figure 2C as showing. I'm assuming curved because the pattern of gray shading forms an arc around the 0,0 point. But then what are the dashed white lines? If they are intended to show model fits of stimuli that elicit the same firing rate, then I think something is wrong with that algorithm because the data do not, to my eye, align with the dashed lines. If they are intended to show theoretically which stimuli would have the same firing rate if the neuron were linear, then I'm confused why the lines are tilted.

The reviewer is correct that the neuron in Figure 2C was fit slightly better by the GQM than the GLM. The GQM vs. GLM comparison for this example cell (quantified by the hyperpixel white noise NLI) is shown in Figure 2D as a red vertical tick mark. This neuron was selected as an example because its NLI lies near the mode of the distribution.

The data shown in Figure 2C are binned for the purposes of visualization. This binning was necessary because stimulus projections onto the two halves of the hyperpixel STA varied continuously. Please see Figure 2 —figure supplement 1 for a scatterplot of the stimulus projections (panel A, red points) and the corresponding binned histogram (panel C). Bins that are far from the center of the box (0,0) contain few stimuli and thus estimates of spiking probability for these bins are noisy. Also, estimates of the probability of spiking are rendered in uncalibrated grayscale, making it difficult to compare differences in gray level by eye. The GLM and GQM were fit to the unbinned data.

We have revised the text to clarify these points and have updated Figure 2C to show firing rate predictions from the GQM fit, which are similar to those from the GLM fit. We note that tilted isoresponse lines are consistent with linearity and indicate that one subfield was dominant over the other.

Results, pg. 10. It seems odd to set the target firing rate to be lower than significantly (substantially) above baseline firing rate for any cells (even three). It would be useful to know the range, mean, and confidence limits around the mean, for the target firing rates (in terms of % above background firing rate).

We have summarized the information about baseline firing rates and target firing rates in Figure 3—figure supplement 2. On average, the target firing rate was 15.65 standard deviations above the mean baseline firing rate. The target firing rate of three neurons fell between the 90^th^ and 95^th^ percentiles (none were lower). Removing these neurons from the data set did not change the main conclusions of this study.

Discussion, pg. 12. I appreciate the point the authors are trying to make, but as framed here, it seems weak ("we used a visual stimulation paradigm that only works if the cells are linear to test if the cells are linear").

We believe that the reviewer is referring to our statement that we "characterized the spatial integration by individual V1 neurons using… white noise RF mapping". White noise stimulation can be used to characterize both linear and nonlinear neurons, but we agree that the concept of an RF map assumes linearity. We have changed the text from "white noise RF mapping" to "white noise visual stimulation".

Discussion, pg. 13. The prior study discussed by the authors concluded that the responses to pairs of bars was linear. Another earlier study (Conway, 2001) showed two-bar interaction maps for two cells, one of which was clearly non-linear (and the non-linear cell was the one with strong S-cone input).

These studies are cited in the revised Discussion.

Discussion, pg. 13. The authors state that the prior work made an assumption that excitatory responses to an increment were the same size but opposite sign to suppression of a contrast decrement. It's not clear to me that this was an assumption rather than a test. The reverse-correlation procedure used in the earlier work used as a baseline the responses to stimuli presented well outside the receptive field, which allows the assumption to be directly tested.

The idea that the excitatory response to a stimulus is the same magnitude, but opposite sign, as the suppression to a contrast-inverted stimulus is described by Conway and Livingstone (2006) as an assumption:

“The suppression shown by many [V1] neurons to some stimuli resulted in a near complete reduction in background firing rate, leading one to suspect that the stimulus was capable of more suppression than could by measured extracellularly. To overcome this rectification nonlinearity in simple cells, the full extent of the response is often inferred by subtracting the responses to opposite-contrast stimuli (Ferster, 1994). The assumption that the excitatory response to a stimulus is the same magnitude, but opposite sign, as the suppression to a stimulus has been shown to be valid for LGN cells in the cat (Martinez et al., 2005). We made this assumption, too, to represent the responses succinctly by difference maps, which allow a direct quantitative calculation of the cone weights.” (emphasis added)

Our intention is simply to paraphrase this description.

Discussion, pg. 16 "In either case, DO cells are a likely basis for the chromatic sensitivity of color-sensitive complex cells (Michael, 1978)." This isn't clear to me. It seems equally plausible (arguably more plausible) that color-sensitive complex cells could derive color tuning by direct sampling of geniculate inputs.

We consider this possibility in the revised Discussion.

Figure 3. It would be helpful to show the STA for each of the example cells. The distribution of data points in Figure 3A is stated in the results as being linear, but it looks curved to me (the two outliers at the far left notwithstanding). None of the example cells in Figure 3 are L-vs.-M cells, which is odd since the majority of "classical" double-opponent cells are defined by L-vs.-M opponency. By contrast, all the cells in Figure 3A could plausibly be simple cells. I'm a bit confused how meaningful the responses of a cell are if they fall on a line that goes through the origin. Responses at the origin are to a null stimulus (blank gray), which indicates that the target firing rate is nota much different from baseline.

The hyperpixel STA for each neuron is presented in Figures 3A–C as the image at the top of each Figure. We have added the pixel white noise STA to these figures as well as bar plots of normalized cone weights (which the color rendering in the figure captures inadequately).

The Reviewer is correct that a point at the origin would indicate a response to a blank screen. None of the data points are at the origin, but some come very close to it and so does the fit. We have added a new supplementary figure (Figure 3 —figure supplement 3) that shows a zoomed in representation of the origin from Figures 3A and 3B.

Reviewer #2 (Recommendations for the authors):General. I think this paper would benefit from a sharpening of terminology and usage throughout. For example, always being explicit about what sort of non-linearity is being considered in any given sentence, and about which of thet two types of white noise experiments is being referred to in each place where a reference is made. I could almost always figure out it what was intended from context, but that was extra work and left some uncertainty, as well distracted from following the flow of ideas. Specific examples given in some of the comments below.

We have edited the manuscript to better indicate which type of non-linearity and which type of white noise are being considered throughout.

Starting on page 3 and throughout. A key to understanding this paper is to understand the taxonomy the authors have applied to the neurons they recorded from. I struggled with the simple cell terminology, though. The criteria the authors use are different from more conventional ways used distinguish simple and complex cells. I think that if one applied F1/F0 criteria using luminance grating stimulation, one might well call at least some of the DO cells simple, as well as possibly some of the cells not studied. The criteria here are that the hyperpixel white noise analysis shows that cone contrast inputs are combined linearly and that the RFs are not cone opponent. I would have an easier time if the term simple cell were not used, to avoid confusion with the conventional taxonomy. Perhaps single opponent cells, or spatially-opponent non-cone-opponent cells. Thesse suggestion don't capture the linearity aspect of the criteria, but nor does the term double opponent and the authors are content with that. Maybe both terms should more clearly denote the linearity criteria. With a different choice of terminology, the relation to the conventional simple/complex taxonomy could be handled entirely in the discussion, with some care devoted to treating possible differences in classification. Avoiding the simple cell terminology might also allow ease exposition on the possible relation between NSNDO cells in the current study and chromatically sensitive complex cells, where I think more elaboration would be helpful.

As pointed out by the reviewer, several criteria had to be met for a neuron to qualify as simple or DO in this study: spatial opponency and the lack of a significant PC1(for both cell types), and cone opponency or cone non-opponency (for DO cells and simple cells, respectively). Capturing all of these criteria with concise labels is difficult. The term "single-opponent" has already been used to describe neurons that have uniform coneopponency throughout their RFs (Johnson et al., 2008; Livingstone 1984; Shapley 2011).

We agree that many of the DO cells that we studied would likely have been classified as simple cells by the standard F1/F0 criterion, which is almost always measured with isochromatic stimuli. Simple cells were first described in the cat, an animal with few cone-opponent V1 cells, and the distinction between cone-opponency and nonopponency is more important in the macaque. Double-opponent cells in macaques are, by definition, cone-opponent, and we know of no widely accepted term for doubleopponent cells that lack cone-opponency. Such cells have RFs consisting of spatially distinct, oriented ON and OFF RF subfields, and are almost certain to be categorized as simple cells by standard criteria. As pointed out by the reviewer, however, so are others.

To maximize the accessibility of our manuscript to the broad readership of *eLife*, we have elected to retain the “simple cell" terminology in our revision. We believe that this label captures the key characteristics of the approximately linear, spatially opponent, cone non-opponent, orientation-tuned cells that we studied. Nevertheless, we have added text explaining that our definition of simple cells is unconventional because is excludes cone-opponent cells and cells with certain types of nonlinearities. We have also revised the manuscript to clarify the categorization criteria, their rationale, and their relationships to classical criteria.

Page 3. Introduction: "The first stage of the primate visual system at which spatial comparison of spectral signals is implemented by individual neurons is area V1." – This statement is mysterious to me. Spatial comparisons are made at the first synapse, as signals from cones interact with signals from horizontal cells, and these interactions can involve spectral differences as signals from different classes of cones participate. Perhaps the authors mean something very specific when the qualify with "by individual neurons" but it is not clear what this is.

Thank you for pointing this out. V1 is the first stage of the primate visual system that contains DO cells. This has been clarified in the revised manuscript.

Page 4. I think more should be said about the 134 neurons that were recorded from and not studied further. These don't have a name in the current study and it is not entirely clear to me what caused them to be excluded. Perhaps only that they did not show any spatial opponency in the initial pixel white noise analysis (page 6), but I'm not sure that's the only reason so more detail on the exclusion criteria would be helpful.

We have expanded discussion of these neurons in the revised manuscript. There are three reasons that neurons were excluded from the current study:

1. The STA was indistinguishable from noise.

2. The STA did not reveal spatial opponency (e.g. it resembled a uniform blob).

3. Electrical isolation was lost early during or prior to *Phase 3* (the isoresponse contour measurement).

Many neurons in the first group (and possibly some in the second group) may have been complex cells. Others may have had contrast thresholds that were too high, RFs that were too small to be adequately stimulated by the pixel white noise, or had noise levels that were too high to produce a clear STA. Neurons in the second group appeared to have spatially homogeneous RFs and were therefore unsuitable for the current study. We were unable to collect sufficient data from neurons in the third group.

Page 5. I agree with the authors that equating stimulus contrast across chromatic directions is a difficult and subtle matter, and that whenever possible it is useful to look for ways to frame questions that are robust to such equation. I diverge in that I don't see that their method avoids this difficulty. Just as one example, the frame selection criteria (pp. 21-22) are based on deciding which STA frame differs most from the background. That calculation is using some metric across color directions. Similarly with the criterion used to decide that the chromatic preferences of the two subfields of a DO neuron differ from each other.

We have edited the text to be clearer about the color space-specificity of the analyses.

As pointed out by the reviewer, the STA frame that differed most from the background was selected on the basis of the (background subtracted) R^2^+G^2^+B^2^, which is a color space-specific statistic. Under the null hypothesis, this statistic has a scaled chi-squared distribution, which simplified the analysis. Had the analysis been done in, for example, (background subtracted) LMS space, correlations between the three color channels would have complicated the distribution (of L^2^+M^2^+S^2^). Correcting for these correlations is equivalent to doing the analysis in RGB space.

For each neuron, there were only 3–6 STA frames that were clearly different from noise. Across these 3–6 frames, R:G:B ratios changed little, so the light selected would have been similar under any reasonable selection rule (e.g. see Figure 2B). The frame selection procedure was used only to select lights on the two sides of the RF to stimulate the cell in *Phase 3*, and we have no reason to think that small changes in these lights would have had a large effect on the results.

As pointed out by the reviewer, Pearson’s correlation coefficient depends on the color space in which the analysis is performed. In the revised manuscript, we present a new analysis of cone weights estimated separately from the two subfields of each STA. This analysis (which is also color space-specific) confirms that chromatic preferences of the simple and DO cells are approximately opposite in this more-physiological color space.

Page 12. While we're on this point, the authors have used device primary RGB as their metric. The justify the reasonableness of this choice by writing "The fact that DO cells and simple cells combined light intensities across space approximately linearly in this representation shows that it is a reasonable one for analyzing V1 neurons." This seems odd, first because it assumes the answer to the question being studied in this paper. Second, because if the combination is revealed to be linear in one choice of color representation, it will also be linear in another choice that is a linear transformation of the first. (Indeed, the authors themselves say this in the sentence after the one quoted.) So I don't see how the linearity of the result tells us much about whether the representational choice was good. If nothing else, I would ask that the authors provide enough information about their device so that others can transform their data into a device independent representation such as LMS contrast with respect to a well-defined choice of cone fundamentals.

We agree and have removed the statement about the reasonableness of RGB space from the text. We provide tabulated primary emission spectra and background RGB levels (γ-corrected) at https://github.com/horwitzlab/Chromatic_spatial_contrast.

Page 6. Figure 2B. I assumed the noise is independent across color channels in RGB, but would be good to say so explicitly.

This information is now stated explicitly.

Page 8. Figure 3. What should we make of the high indices for some DO cells, and the low indices for many NSNDO cells? Both distributions suggest at least that possibility that neither group is a homogeneous population with respect to non-linear spatial summation. More discussion of this seems worthwhile.

We agree and have amended the Discussion to clarify our perspective that DO cells lie on a continuum with some NSNDO cells with regard to the linearity of spatial summation.

Page 10. How were the target firing rates chosen?

Baseline firing rates for each neuron were monitored online during *Phases 1 and 2* of the experiment. Target firing rates were chosen such that they were substantially higher than the mean baseline firing rate (on average, 15.65 SDs higher). We have added a supplementary figure (Figure 3 —figure supplement 2) showing target firing rates and baseline firing rates for each neuron studied.

Page 13. "Nearly all quantitative studies of V1 RFs have used achromatic stimuli". Perhaps worth listing the studies of this sort that have used chromatic stimuli. Also, I think that by "V1 RFs" the authors mean "V1 spatial RF properties" and this might be worth spelling out.

We have revised the text as per the reviewer’s suggestions.

Page 13. "This maximal response exceeded the response to either bar in isolation, consistent with linearity as well as with other models." Might make explicit that the linearity in this sentence is the linearity of spatial summation, as well as that the reason the result could be consistent with summation linearity is that there could be a response non-linearity after spatial summation.

We have revised the text as per the reviewer’s suggestion.

Page 14. "Reclassifying cells …. " There's a lot of cell classification in this paper; perhaps sharpen here to emphasize that it is the simple/DO dichotomy classification being referred to here.

We have revised the text as per the reviewer’s suggestion.

Page 14. "Most other recent studies of DO cells used cone-isolating stimuli, which cannot reveal interactions among cone types (Conway, 2001; Johnson et al., 2001, 2004; Conway and Livingstone, 2006; Johnson et al., 2008)." Johnson et al., used mixtures in some of their reported measurements.

We have revised the sentence.

Page 14. "We classified neurons with nonlinear responses to white noise as NSNDO. This criterion was necessary to satisfy the assumptions underlying the conversion of the STA to cone weights." (A) In the first sentence, "nonlinear" seems too general since the paper believes it worthwhile to check for a specific type of nonlinearity in the simple and DO cells. (B) I don't understand this because the authors go on to apply the conversion of the STA to cone weights with the NSNDO cells, in that such conversion is used for the test in Phase 2, as shown in Figure 3F. (C) Here is an other places, it would be helpful to say "hyperpixel white noise" rather than just "white noise"

A. We have revised this part of the text to clarify that we mean specifically nonlinearities that manifest in a PC1 that is significantly larger than expected by chance (e.g. the contrast-polarity invariance exhibited by complex cells).

B. We use the STA of the NSNDO cells only to identify lights on the two sides of the RF that are likely to drive a strong response. We agree that calculating cone weights from the NSNDO cells is unlikely to be a useful conversion. Nevertheless, we do this in Figure 6 for the sake of completeness.

C. We have specified which type of white noise we are referring to throughout the revised manuscript.

Page 15. "Pooling LGN afferents with the same sign (ON or OFF) creates non-opponent spectral sensitivity." I'm not following what is meant here, need more detail on what is being assumed about chromatic properties of the afferents as well as exactly what is being pooled.

Thank you for pointing out this confusing section. The intended meaning is that pooling L- and M-cone-dominated ON afferents from one part of the visual space produces sensitivity to cone non-opponent increments, the precise details of which depends on the net spectral sensitivity of the afferents (e.g. L-cone dominated, M-cone dominated, or balanced). Surrounds are assumed to contribute little, spatially, relative to the center, and this is the case in the LGN under conditions similar to those examined in this study (e.g. (Horwitz 2020)). On the other hand, S-ON and S-OFF afferents carry strong, asymmetric L- and M-cone signals and therefore do not fit into this simple framework. This is explained more completely in the revised manuscript.

Page 17 and Figure 4. If the authors are going to show this data can they say anything about how the results for the experiments that are the focus of the current paper would come out for these two cells? That would connect these preliminary data better to the rest of the paper. And, did the authors find any cells like the one shown in 4C, but with the responses to luminance and chromatic gratings flipped?

Thank you for this question. We suspect, but do not know, that the neurons in Figure 7C (formerly Figure 4C) would have been either passed over in this study or would have been classified as NSNDO given their insensitivity to the phase of drifting gratings.

No, we have never seen a complex cell with a vigorous response to chromatic (L-M) modulation that did not also have a vigorous response to luminance (L+M) modulation (see also Horwitz, Chichilnisky, and Albright, 2007).

Page 22. Please provide criteria used to decide whether the two subfields had distinct chromatic preferences, and how this related to cell classification. On page 7, correlations (computed how?) between the two halves of the STA are given, and it sounds like this is result obtained before selecting or classifying cells. One page 24 a percent variance explained number is given, again independent of cell classification. But I'm not entirely sure that this property wasn't required for cells to be included for further study.

The criteria used to decide whether the two subfields had distinct chromatic preferences online were qualitative. The pixel STA was rendered on the display of the data acquisition computer. If the experimenter could identify two visually distinct regions of the STA, these regions were manually segmented for the hyperpixel white noise stimulus (*Phase 2*). As soon as the two subfields of the hyperpixel STA (again, rendered online) became visually distinct and clearly different from the background gray, the experiment advanced to *Phase 3*.

The % variance explained refers to how well the hyperpixel STA is recovered by combining the color and spatial weighting functions computed by SVD. All cells were subjected to this analysis. Every neuron in the data set had a spatial weighting function consisting of one positive and one negative weight, consistent with opposite spectral sensitivity in the adjacent subfields.

Cells that survived this filter were classified into simple, DO, and OSO categories. We computed Pearson’s correlation between the 2 sets of background-subtracted RGB values (one for each subfield) from the hyperpixel STA of each neuron. We have revised the text to explain this procedure more clearly.

Page 22. The authors note that using small pixels and a fast frame rate in the white noise experiments leads to the neurons being weakly driven. They might comment/think about whether the choices they made for these parameters are optimal – if they were to do the work again, could they consider using larger white noise pixels and a slower refresh rate to obtain better responses? That is, small/fast is at one end of a continuum, and maybe some other choice would be better for future work of this sort. Similarly, any requirement for use of the Gaussian distribution on the noise? Maybe a uniform or some other distribution would produce larger drive.

The pixel size was matched to the size of the average fixational saccade (0.2°) made by monkeys viewing similar displays (Hass and Horwitz 2011). Longer frame durations (or, really, repeated frames, since the stimulus was shown on a CRT) might have been preferable for this study because temporal aspects of the response were of secondary interest. Gaussian distributions facilitate the interpretation of the spike-triggered covariance analysis relative to uniform distributions, which we agree would likely drive the cells harder (Paninski 2003).

Page 22, last paragraph and into p. 23. The paragraph starts off talking about frames as six-dimensional vectors, but by the end it sounds like two separate 3-dimensional projections are done, yielding the two scalars one for each subfield. Would be clearer, I think, if only one of the two ways of talking about it is used here. Also might remind reader that this is the hyperpixel white noise STA, not the pixel white noise STA.

Thank you for pointing out this confusing passage. We have simplified the explanation and emphasized that the analysis was performed on two separate 3-dimensional projections onto the hyperpixel STA.

Page 23. We visualized a firing rate map from these projections by computing the ratio of spike-triggered stimuli to the total stimuli." I don't understand what metric of stimuli is being used in the ratio, which visualization in the paper this refers to, nor how the ratio of stimuli as described provides a map of firing rate.

Thank you for alerting us to this confusing section. The stimuli were represented as linearized, background-subtracted RGB values, and these stimuli were projected onto the two halves of the STA. Each stimulus therefore produced a pair of scalars that define a location in a plane (Figure 2C). Because the elements of the white noise stimulus are jointly Gaussian, this distribution of projections is a bivariate Gaussian.

This procedure was performed on every stimulus that was shown in the experiment. Some of these stimuli preceded spikes, and most did not. We gridded the plane into small, square bins and, within each bin, counted the number of stimuli that preceded a spike and the number that did not. We estimate the probability of spiking, conditional on a stimulus being in a particular bin, as the number of stimuli in that bin that preceded a spike divided by the total number of stimuli in that bin. We refer to the probability of spiking as a function of two stimulus projections as a firing rate map. This has been clarified in the revised text.

Page 23. "Direction represents the overall contrast between …" Worth noting that what these axes represent in stimulus space is cell-specific, in that the axes for each cell are derived from that cell's hyperpixel STA. Also, although this is a matter of convention and up to the authors, I would find it more intuitive if the STA axes were the x and y axes of the plot, not the 45 and 135 directions.

We have revised the text based on reviewer’s suggestions.

The rationale for the unusual plotting convention is that most of the stimuli that were sampled in *Phase 3* provided positive contrast to both subfields. If we were to align the STA axes with the abscissa and ordinate, the figure would be rotated by 45°, and approximately half of the figure would be unused white space. In an effort to use the figure space efficiently, we have elected to retain the current rotation of the axes.

Page 24. 232 neurons were recorded from. 98 are reported on here. What can we say about the rest, and why they were not included here. The one clear exclusion criterion is that some neurons had only a single spatial region revealed by the pixel white noise STA. Was that the only reason? On this point, I don't think NSNDO is the best term for the set of neurons out of the 98 that were not classified as simple or DO, because of all the other neurons that were not included and that also weren't classified as simple or DO.

Please see our response to comment #26 for the answer to this question. We refer to the NSNDO cells as “other spatially opponent” (OSO) cells in the revised manuscript.

Page 24. Is it a theorem that if you take the first row singular vector and the first column singular vector, that this is the optimal color-space separable accounting of the STA? Or could a joint analysis do better?

Yes, the Eckart-Young-Mirsky theorem (Eckart 1936) states that the outer-product of the first column singular vector and the first row singular vector (multiplied by the first singular value) is the separable (rank 1) matrix that comes as close as possible to the original matrix in the sense of Frobenius norm. We have added a sentence to the text to inform the reader of the optimality of this procedure.

Page 24-25. On page 25, the classification of cells into simple, DO, and NSNDO is described. Because the term "white noise" is used to describe the protocol used for the classification and because this applies to both Phase 1 (pixel white noise) and Phase 2 (hyperpixel white noise) of the experiment, it would be good to remind the reader that it is (I think) the hyperpixel white noise that's relevant here. Indeed, I would try to be explicit each time in the paper the term white noise or STA is used, which of the two it refers to. Also, why is a substantial S cone input a requirement to be a DO cell? This seems like it would exclude cells with perfectly good spatially opponent L-M inputs. And, I'm not clear what happens to such cells, if there were any. I think from the next sentence that like neurons with a significant PC1, they get called NSNDO. But because the flow of this section had me thinking at the start that it was just those neurons with signifiant PC1 that were NSNDO, I'm also not clear on this. If they are NSNDO, then it is worth being explicit elsewhere that heterogeneous criteria were used to place neurons in this category, and perhaps looking at NLI separately for cells that arrived as NSNDO viat different paths.

We apologize for the ambiguity and explicitly mention which white noise (pixel or hyperpixel) was used throughout the revised text.

Our criteria for double-opponency did admit neurons with spatially opponent L-M inputs and small (or even non-existent) S-cone inputs. If a neuron had oppositely signed L- and M-cone weights, each of which accounted for at least 20% of the total, the neuron was classified as double-opponent, irrespective of the magnitude of the S-cone weight. The threshold on weight magnitude prevented neurons with non-opponent cone weights from being misclassified (small, independent noise in RGB produces small, anticorrelated noise in LM). Additionally, neurons were classified as DO if they had oppositely signed cone weights and an S-cone weight that accounted for at least 20% total. The former criteria selected the opponent L-M DO cells whereas the latter criteria selected the S-M+L , S+M-L and S-M-L DO cells (irrespective of the magnitudes of their L- and M-cone weights). We have rewritten the text to clarify the DO cell classification better.

The revised text is more explicit about the multiple criteria that were used to place neurons in the OSO (formerly NSNDO) category. Large differences across neurons that arrived in this category via different routes were looked for in response to this comment but not observed. (75% of the cells in the OSO category arrived via the PC1 criterion.)

Page 26. I'm surprised that Equation 1 doesn't include an amplitude parameter that captures the overall responsivity of the cell.

The amplitude parameter in Equation 1 is exp(c), where "c" is a fitted parameter. “Pred resp” in Equation 1 represents the probability that a stimulus will evoke a spike, and therefore has an upper asymptote of 1.

Figure 2 —figure supplement 1. I am not sure what is in panel D. Are these from Phase 2? And how does the 5th and 95th percentile information mentioned in the caption show up in the figure? More generally, I think the point of this figure is to provide a sense of how much harder the Phase 3 stimuli drive the cells than the Phase 1 and 2 stimuli, but I am not seeing this in the figure itself. Where are the Phase 3 data?

Thank you for alerting us to this confusing figure. Panel D shows data from *Phase 2* in the same format as panel A. The square firing rate map in the center of panel D shows, in grayscale, the proportion of stimuli (the red dots in panel A) that drove spikes. The 5th and 95th percentiles relate to the size of this square; had we included stimuli from the 1st to 99th percentiles, the box would have subsumed nearly all of the red points in panel A and thus been larger, but our estimate of the proportion of stimuli that drove a spike in bins near the edges would have been noisier because counts in these bins would have been low.

The previously submitted manuscript did not include an analysis of how much harder the *Phase 3* stimuli drove the cells than the *Phase 1 and 2* stimuli. We include this new analysis in Figure 2 —figure supplement 2 of the revised manuscript.

Figure 2 —figure supplement 2. Can you clarify the temporal parameters that go into this analysis?

We considered 15 frames of the hyperpixel STA prior to when a spike occurred, which is equivalent to 200 ms (frame rate = 75 Hz)(Figure 2B). We then convolved the hyperpixel white noise video with the two halves of the hyperpixel STA to obtain the projection values for each time bin. We have clarified these temporal parameters in the revised text.

Figure 3 —figure supplement 3. The staircase shown suggests that using the last stimulus presented does not give as good an estimate of stimulus that produces the criterion firing rate as would an analysis of the whole staircase. Do I understand that the last stimulus magnitude is what was used? If so, please elaborate why. If not, please clarify.

Yes, the final stimulus contrast magnitude was used to estimate the isoresponse point. Inspired by this comment, we repeated the analysis, defining isoresponse points in two new ways: first, as the geometric mean of the last 5 staircase points (geometric mean because the contrast was scaled multiplicatively in the staircase), and second, interpolation using a Naka-Rushton fit to the entire staircase assuming Poisson error. As shows in Author response image 2 and Author response image 3 the results from these three procedures are almost identical. For simplicity, we have decided to stay with the original definition.

**Author response image 2. sa2fig2:** Comparison of staircase termination points with the geometric mean of the last 5 points in the staircase. (**A**) Data from the example DO cell shown in Figure 3A. Black dots indicate staircase terminations and red dots indicate the geometric mean of the last 5 points in the staircase. (**B**) Staircase termination points are plotted against the geometric mean of the last 5 points of successful staircase terminations. (**C** and **D**) Same as **A** and **B**. but for the example simple cell shown in Figure 3B. (**E** and **F**) Same as **A** and **B**. but for the example OSO cell shown in Figure 3C.

**Author response image 3. sa2fig3:** Comparison of staircase termination points with contrasts producing the target firing rate as interpolated from fitted Naka-Rushton functions. (**A**) Data from the example DO cell shown in Figure 3A. Black dots indicate staircase terminations and red dots indicate Naka-Rushton fitted points. (**B**) Staircase termination points are plotted against the Naka Rushton fitted points. (**C**) and (**D**) same as (**A**) and (**B**) but for the example simple cell shown in Figure 3B. (**E**) and (**F**) same as (**A**) and (**B**) but for the example OSO cell shown in Figure 3C.

References

Eckart C, and Young, G. The approximation of one matrix by another of lower rank. *Psychometrika* 1: 211-218, 1936.

Hass CA, and Horwitz GD. Effects of microsaccades on contrast detection and V1 responses in macaques. *J Vis* 11: 1-17, 2011.

Horwitz GD. Temporal information loss in the macaque early visual system. *PLoS Biology* 18: e3000570, 2020.

Horwitz GD, and Hass CA. Nonlinear analysis of macaque V1 color tuning reveals cardinal directions for cortical color processing. *Nature Neuroscience* 15: 913-919, 2012.

Johnson EN, Hawken MJ, and Shapley R. The orientation selectivity of color-responsive neurons in macaque V1. *The Journal of Neuroscience* 28: 8096-8106, 2008.

Livingstone MS, Hubel, D. H. Anatomy and physiology of a color system in the primate visual cortex. *J Neurosci* 4: 309-356, 1984.

Paninski L. Convergence properties of some spike-triggered analysis techniques. *Advances in neural information processing systems* 189-196, 2003.

Shapley R, and Hawken, M. J. Color in the cortex: single-and double-opponent cells. *Vision research* 51: 701-717, 2011.